# A green lifetime biosensor for calcium that remains bright over its full dynamic range

Franka H van der Linden[1], Stephen C Thornquist[2], Rick M ter Beek[1], Jelle Y Huijts[1], Mark A Hink[1], Theodorus W J Gadella[1], Gaby Maimon[2], Joachim Goedhart[1]*

[1]Swammerdam Institute for Life Sciences, Section of Molecular Cytology, van Leeuwenhoek Centre for Advanced Microscopy, University of Amsterdam, Amsterdam, Netherlands; [2]Laboratory of Integrative Brain Function and Howard Hughes Medical Institute, The Rockefeller University, New York, United States

**\*For correspondence:**
j.goedhart@uva.nl

**Competing interest:** The authors declare that no competing interests exist.

## eLife Assessment

This manuscript reports on an FLIM-based calcium biosensor, G-CaFLITS. It represents an **important** contribution to the field of genetically-encoded fluorescent biosensors, and will serve as a practical tool for the FLIM imaging community. The paper provides **convincing** evidence of G-Ca-FLITS's photophysical properties and its advantages over previous biosensors such as Tq-Ca-FLITS. Although the benefits of G-Ca-FLITS over Tq-Ca-FLITS are limited by the relatively small wavelength shift, it presents some advantages in terms of compatibility with available instrumentation and brightness consistency.

**Abstract** Fluorescent biosensors toggle between two states, and for the vast majority of biosensors, one state is bright and the other state is dim. As a consequence, there is a substantial difference in the signal-to-noise ratio (SNR) for the two states. The dim state has a low SNR, which is problematic when precise, quantitative measurements are needed. During the engineering of a red-shifted variant of an mTurquoise-based calcium sensor, we serendipitously generated a green-emitting sensor that shows high brightness in both the calcium-bound and -unbound state, while still showing a calcium-dependent lifetime change of >1 ns. This sensor, named G-Ca-FLITS, is comparable in brightness to the bright state of GCaMP3 and jGCaMP7c in mammalian cells. The calcium-induced loss in fluorescence intensity is only around 30% and therefore we observe little variation in the SNR when calcium levels change. G-Ca-FLITS shows negligible sensitivity to pH in the physiological range, like its turquoise parent. Using fluorescence lifetime imaging (FLIM), we measured the calcium concentration with G-Ca-FLITS in various organelles and observed in HeLa cells transient and spatially heterogeneous calcium elevations in mitochondria. Finally, we evaluated the use of G-Ca-FLITS and its turquoise predecessor for two-photon FLIM in *Drosophila* brains.

## Introduction

Genetically Encoded Calcium Indicators (GECIs) are by far the largest family of biosensors, with over 100 members (*Greenwald et al., 2018*) including the well-known GCaMP series. Many GECIs harboring a single fluorescent protein (FP), like GCaMPs, are optimized for a large intensity change and have a (very) dim state when calcium levels are below the $K_D$ of the probe (*Akerboom et al., 2013*; *Dana et al., 2019*; *Shen et al., 2018*; *Zhang et al., 2023*; *Zhao et al., 2011*). This feature is advantageous

when using the probe as a binary detector for increases in $Ca^{2+}$ concentrations. However, it makes robust quantification of low, or even intermediate, calcium concentrations extremely challenging.

To enable more precise measurements of low calcium concentrations, a biosensor with an inverted response (*Barykina et al., 2016*) is a better alternative, as these biosensors are bright in the calcium-free state. However, this only partially addresses the issue, as the measurement precision for high calcium concentrations is compromised since the intensity is low in the calcium-saturated state.

Another disadvantage for quantification with intensity-based biosensors is that the fluorescence intensity is affected by many factors (*Culley et al., 2024*; *Waters, 2009*), complicating the measurement of absolute calcium concentrations with biosensors that use intensity contrast as a read-out. Therefore, we and others have engineered genetically encoded biosensors that use other read-outs. New biosensors that report with a change in photochromic behavior (*Bierbuesse et al., 2022*), anisotropy (*Laskaratou et al., 2021*), or fluorescence lifetime (*Díaz-García et al., 2017*; *Klarenbeek et al., 2015*; *van der Linden et al., 2021*) have been developed.

Despite the more robust read-outs, the measurement precision of all approaches depends on the number of detected photons. For instance, the turquoise lifetime sensor Tq-Ca-FLITS that we developed (*van der Linden et al., 2021*) shows a threefold intensity change. Yet, the ideal lifetime biosensor would be bright in both states, enabling high measurement precision across its entire dynamic range. Such a biosensor could be used for imaging low concentrations of calcium and subsequent increases, for example, in organelles of mammalian cells, including mitochondria (*Parry et al., 2024*).

Calcium signaling in mitochondria is reported to be linked with several pathophysiological processes including inflammatory responses, diabetes, heart failure, neurological diseases, and cancer (*Delierneux et al., 2020*; *Giorgi et al., 2018*). Over the years, mitochondrial calcium concentrations have been measured with imaging techniques; however, there are discrepancies between the reported concentrations. Accurate determination of the affinity of a sensor is important and there are several issues that need to be considered during the calibration and the measurements: (1) the concentrations can only be measured with sufficient precision when they are in the range between $10 \times K_D$ and $1/10 \times K_D$, (2) the calibration is only valid when the two extremes are reached during the calibration procedure (*Fernandez-Sanz et al., 2019*), (3) the sensor's kinetics should be sufficiently fast enough to be able to track the calcium changes, and (4) the biosensor should be compatible with the high mitochondrial pH of 8 (*Abad et al., 2004*; *Llopis et al., 1998*).

Clearly, there is a need for robust biosensors that satisfy all criteria discussed above, enabling quantitative imaging of intracellular calcium concentrations. In this paper, we document the development of a green calcium sensor by an optimized bacterial screening method. The biosensor displays a strong lifetime contrast and at the same time remains bright in the calcium-free and -saturated states. We report the spectroscopic properties of the purified sensor and show its use for robust measurements of calcium levels in mitochondria of HeLa cells and endothelial cells by lifetime imaging.

**Table 1.** Lifetime and spectral screen of candidate sensors to create G-Ca-FLITS.
Sensor variants are indicated by their mutations with respect to Tq-Ca-FLITS. The calcium-bound state (sat) is measured in presence of 0.1 mM $CaCl_2$ and the calcium-free state (apo) after addition of 9.5 mM EDTA.

| Sensor variant | Excitation max. (nm)* | | Emission max. (nm)* | | Phase lifetime (ns)[†] | | |
|---|---|---|---|---|---|---|---|
| | apo | sat | apo | sat | apo | sat | Change |
| Tq-Ca-FLITS | 437, 455 | 437, 456 | 492, 502 | 480, 507 | 1.72 | 2.86 | 1.14 |
| T203F | 459 | 458 | 506 | 507 | 1.71 | 2.19 | 0.48 |
| T203H | No red shift | No red shift | No red shift | No red shift | 2.82 | 2.90 | 0.08 |
| T203Y | 461, 488 | 459, 478 | 516 | 514 | 3.08 | 2.20 | −0.88 |
| I167F, T203Y | 460, shoulder 480 | 460, 479 | 512, shoulder 490 | 516 | 2.59 | 2.88 | 0.29 |
| I167H, T203Y | 459, 478 | 460, 479 | 514 | 519 | 2.92 | 3.11 | 0.18 |
| I167F | No red shift | No red shift | No red shift | No red shift | 1.52 | 2.17 | 0.65 |

*Emission and excitation maxima are only indicated if a red shift of the spectrum with respect to Tq-Ca-FLITS was observed.

[†]Phase lifetimes were measured at 37°C by FD-FLIM. The lifetime change is calculated as the lifetime in the calcium-bound state minus the calcium-free state.

## Results

### Creating a red-shifted calcium sensor with lifetime contrast

We aimed to generate a green biosensor with lifetime contrast. Therefore, we used the previously developed turquoise sensor (Tq-Ca-FLITS) and mutated residues that potentially induce a red shift. We subjected the Tq-Ca-FLITS to directed mutagenesis of residues T203 (mTurquoise2 numbering, T81 in the full sensor) and I167 (mTurquoise2 numbering, I45 in the full sensor) to aromatic amino acids. Both residues are located close to the chromophore. The mutation T203Y is known to cause a red shift in green fluorescent proteins (*Ormö et al., 1996*) and mTurquoise2 (*Gorbachev et al., 2017*). We anticipated that π–π stacking of the aromatic residue with the chromophore would result in the color shift in Tq-Ca-FLITS as well. The candidate sensors were expressed in bacteria, and the bacteria were subjected to the optimized bacterial lysis protocol (*Table 1*, Appendix 1).

Mutations T203Y and T203F both resulted in a red-shifted excitation and emission spectrum in both calcium states, but the shift was more pronounced for the T203Y mutation. This Tq-Ca-FLITS_T203Y variant also showed a large phase lifetime contrast of $\Delta\tau_\varphi = -0.9$ ns. Surprisingly, this change is inverted compared to Tq-Ca-FLITS ($\Delta\tau_\varphi = 1.1$ ns). The T203F mutation also inverted the lifetime response; however, the contrast was about half of the Tq-Ca-FLITS_T203Y variant. A mutation of I167F or I167H in the background of the T203Y variant, or a separate I167F mutation on Tq-Ca-FLITS, reduced the lifetime contrast and did not result in a (further) red shift.

We also tested the T203Y and T203H mutations on different circular permutations (cp) of Tq-Ca-FLITS, which we created during the development of the turquoise sensor (*van der Linden et al., 2021*). In almost all cp variants with the T203Y mutation, the same red shift in both the excitation and emission spectra was observed as for Tq-Ca-FLITS_T203Y. An exception was the cp150-T203Y variant, where we observed a red shift only for the calcium-bound state in both excitation and emission spectra (*Supplementary file 1*, *Appendix 1—figure 3*). This variant differs from Tq-Ca-FLITS_T203Y by only a F146Y mutation. A T203H mutation did not cause a red shift in any of the cp variants.

### Improving the green lifetime sensor by mutagenesis

We aimed to further improve Tq-Ca-FLITS_T203Y by mutagenesis in the regions linking the calcium-binding domains to the FP. Using a PCR approach, we created a library of sensors with any possible amino acid on positions V27 and N271 (numbering according to the sensor, see *Figure 1—figure supplement 1*). Bright green fluorescent colonies on an agar plate were selected for the bacterial lysis test. We found three sensors with an improved lifetime contrast: with alanine or valine on position 27,

**Table 2.** Lifetime contrast of improved green versions of Tq-Ca-FLITS.

| | | Bacterial lysate[†] | | Response in HeLa cells[‡] | | | | | | |
| | | Phase lifetime (ns) | | Phase lifetime (ns) | | | Modulation lifetime (ns) | | | |
| Sensor variant* | State | Mean | Change | Mean | sd | Change | Mean | sd | Change | *n* |
|---|---|---|---|---|---|---|---|---|---|---|
| G-Ca-FLITS (Tq-Ca-FLITS_T203Y-AD) | apo | 3.83 | | 3.19 | 0.08 | | 3.65 | 0.07 | | 314 |
| | sat | 2.45 | −1.37 | 2.02 | 0.06 | −1.17 | 2.50 | 0.08 | −1.15 | 267 |
| Tq-Ca-FLITS_T203Y-VS | apo | 3.39 | | 2.90 | 0.06 | | 3.37 | 0.07 | | 150 |
| | sat | 2.15 | −1.23 | 1.98 | 0.07 | −0.92 | 2.42 | 0.08 | −0.95 | 116 |
| Tq-Ca-FLITS_T203Y-AS | apo | 3.85 | | 3.19 | 0.08 | | 3.63 | 0.07 | | 306 |
| | sat | 2.62 | −1.23 | 2.23 | 0.06 | −0.96 | 2.74 | 0.06 | −0.89 | 345 |
| Tq-Ca-FLITS_T203Y | apo | 3.34 | | 2.95 | 0.08 | | 3.42 | 0.08 | | 193 |
| | sat | 2.21 | −1.13 | 2.13 | 0.06 | −0.82 | 2.57 | 0.07 | −0.85 | 201 |

*'AD' stands for the mutations V27A and N271D, 'VS' for V27 and N271S, and 'AS' for V27A and N271S.

[†]The fluorescence lifetime of bacterial lysates was measured at RT by FD-FLIM after addition of 0.1 mM $CaCl_2$ and after addition of 9.5 mM EDTA. Mean phase lifetimes of the full field of view of the microscope are indicated.

[‡]The fluorescence lifetime of HeLa cells expressing the different sensor variants was measured without stimulation and after addition of 5 µg/ml ionomycin and 5 mM calcium to the medium, measured at 37°C. The mean and standard deviation (sd) of the fluorescence lifetime of all cells (*n*) are indicated.

**Table 3.** Photophysical properties of G-Ca-FLITS and intermediate variant.

Apo – calcium-free state (10 mM EGTA), sat – calcium-bound state (39 µM free Ca²⁺), ε – extinction coefficient, QY – quantum yield with 95% confidence interval between curly brackets.

| | State | $\lambda_{abs}$ (nm) | $\lambda_{em}$ (nm) | $\varepsilon_{max}$ (M⁻¹ cm⁻¹) | QY {CI} (%) |
|---|---|---|---|---|---|
| | apo | 474 | 515 | 29,300 | 41.1 {40.5–41.7} |
| G-Ca-FLITS | sat | 476 | 517 | 30,900 | 25.9 {25.4–26.5} |
| | apo | 470 | 516 | 28,900 | 43.8 {43.2–44.3} |
| Tq-Ca-FLITS_T203Y | sat | 475 | 518 | 30,900 | 30.7 {30.4–31.1} |
| mTq2_T203Y | | 471 | 514 | 27,400 | 23.8 {23.5–24.2} |

and aspartic acid or serine at position 271 (**Table 2**). The sensor with V27A and N271D gave the best results with a contrast of $\Delta\tau_\varphi = -1.4$ ns. Notably, the lifetime contrast of Tq-Ca-FLITS_T203Y is larger than measured in the previous screen (**Table 1**), which is due to a difference in temperature: the first screen was performed at 37°C while this second one was performed at room temperature.

Next, we measured the lifetime response of the three improved variants in HeLa cells (**Table 2**), before and after saturation of the sensor with ionomycin and calcium. Again, the variant with V27A and N271D gave the best results, with a lifetime change of –1.2 ns (both $\Delta\tau_\varphi$ and $\Delta\tau_M$, high calcium minus low calcium), which is an improvement of ~0.3 ns in lifetime change with respect to the parental sensor Tq-Ca-FLITs. We decided to name the new lifetime sensor G-Ca-FLITS, for **G**reen **Ca**lcium **F**luorescence **LI**fe**T**ime **S**ensor.

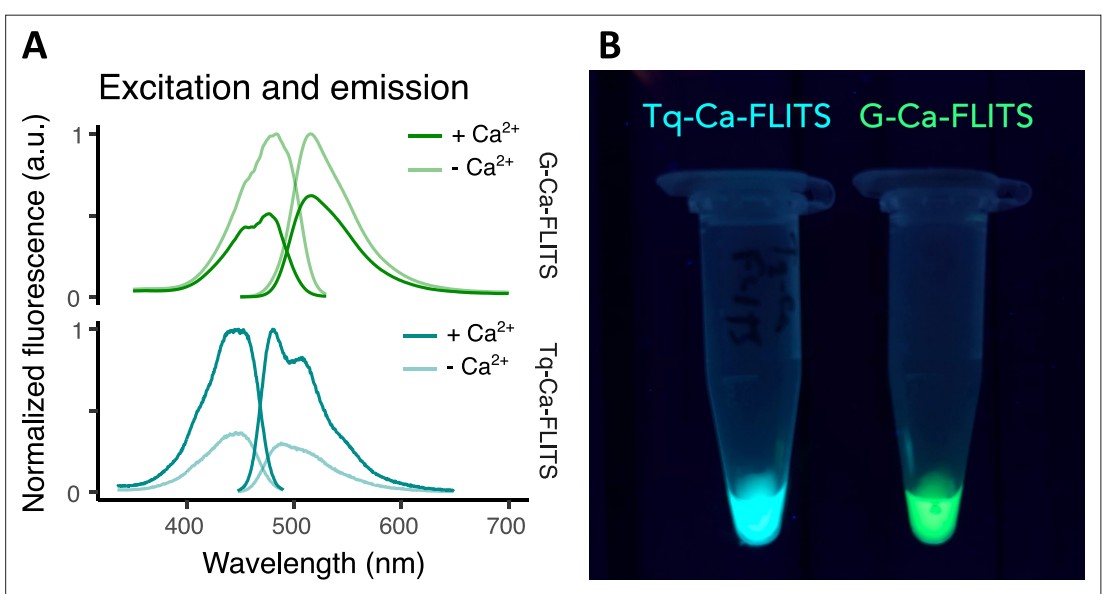

**Figure 1.** The excitation and emission peaks of G-Ca-FLITS are red-shifted relative to Tq-Ca-FLITS. (**A**) The excitation and emission spectra show in lighter lines the calcium-free state (10 mM EGTA) and the darker lines the calcium-bound state (39 µM free Ca²⁺). The Tq-Ca-FLITS is shown as a reference and was reported before (**van der Linden et al., 2021**). Spectra are normalized to the maximum of the calcium-free state for G-Ca-FLITS. (**B**) The photo shows the fluorescence from tubes of purified biosensors when excited with UV (312 nm).

The online version of this article includes the following source data and figure supplement(s) for figure 1:

**Source data 1.** Source data for *Figure 1A*: excitation and emission spectra.

**Figure supplement 1.** Sequence alignment of Tq-Ca-FLITS and G-Ca-FLITS.

**Figure supplement 2.** Excitation and emission spectra of Tq-Ca-FLITS_T203Y and mTq2_T203Y.

**Figure supplement 3.** Quantum yield (QY) determination.

**Figure supplement 4.** Determination of the extinction coefficient by unfolding of the proteins.

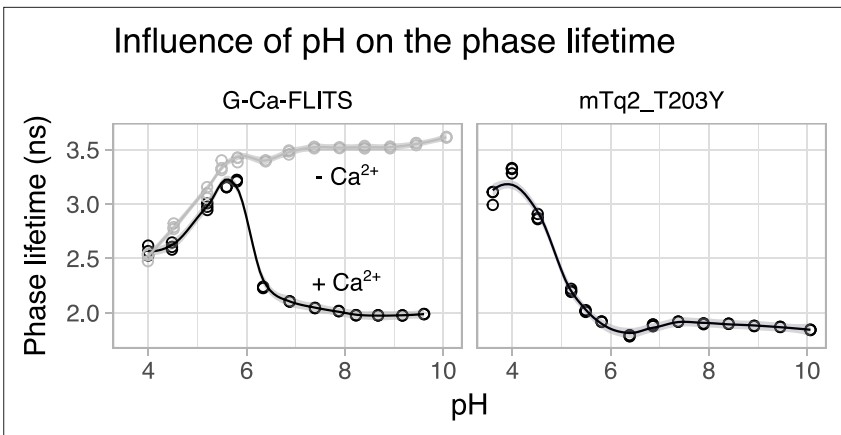

**Figure 2.** Influence of pH on phase lifetime of the proteins. Fluorescence lifetime of proteins diluted in pH buffer was measured ($n = 3$). In the case of G-Ca-FLITS, this was done in the presence (gray, 0.1 mM $CaCl_2$) or absence of calcium (black, 5 mM EGTA). A smooth curve is fitted through the data using the loess method, using $\alpha = 0.4$. The gray band indicates the 95% confidence interval of the smooth fit.

The online version of this article includes the following source data and figure supplement(s) for figure 2:

**Source data 1.** Source data for *Figure 2*.

**Figure supplement 1.** Influence of pH on modulation lifetime of the proteins.

## Characterization of G-Ca-FLITS

The G-Ca-FLITS has three mutations relative to its parent Tq-Ca-FLITS (*Figure 1—figure supplement 1*). In order to characterize its biophysical properties, we isolated G-Ca-FLITS from bacteria and purified the protein by a Ni-NTA column. The intermediate sensor Tq-Ca-FLITS_T203Y and a mTurquoise2 with T203Y mutation (mTq2_T203Y) were also purified for comparison (*Table 3*, *Figure 1—figure supplement 2*). G-Ca-FLITS shows excitation and emission spectra that are evidently red-shifted when compared to the Turquoise variant (*Figure 1*).

We determined the quantum yield (QY) of G-Ca-FLITS to be 41% for the calcium-free state, and 26% for the calcium-bound state (*Table 3*, *Figure 1—figure supplement 3*). This difference of 15% is slightly higher than in the intermediate variant Tq-Ca-FLITS_T203Y (13%). Interestingly, the QY of mTq2_T203Y is 24%, comparable to the calcium-bound state of G-Ca-FLITS.

For G-Ca-FLITS, the extinction coefficient is for both states around 30,000 $M^{-1}$ $cm^{-1}$, which is slightly higher than mTq2_T203Y (*Table 3*, *Figure 1—figure supplement 4*).

We measured the pH sensitivity of G-Ca-FLITS and mTq2_T203Y (*Figure 2*, *Figure 2—figure supplement 1*). In the calcium-free state, the G-Ca-FLITS shows an increase in fluorescence lifetime in the range pH 4–6, followed by a stable lifetime above pH = 6. In contrast, in the calcium-bound state, the lifetime increases until pH = 5.5, followed by a drop and a stable lifetime above pH = 7. The mTq2_T203Y protein also shows a maximum of fluorescence lifetime at a low pH (pH = 4) followed by a drop and a stable lifetime above pH = 6.

The calcium affinity of the lifetime read-out of G-Ca-FLITS was determined at 37°C. To this end, we measured the lifetime of the sensor in a range of calcium buffers using frequency-domain FLIM (FD-FLIM) and plotted this in a polar space (*Figure 3A*, *Figure 3—figure supplement 1*). The two extreme values (zero calcium and 39 µM free calcium) are located on different coordinates in the polar plot, and all intermediate concentrations are located on a straight line between these two extremes. Based on the position in the polar plot, we determined the fraction of sensor in the calcium-bound state, while considering the intensity contribution of both states (*Figure 3B*).

The intensity contribution was determined based on the excitation and emission spectra, the extinction coefficient per wavelength, the QY and the properties of the microscope, i.e. the excitation/emission filters, the excitation source and the detector sensitivity profile (Appendix 2, *Appendix 2—figure 1*). A calibration curve was fitted to the fraction of sensors in the calcium-bound state for each concentration, from which the calcium affinity was determined to be $K_D = 209$ nM (*Table 4*). Using the variation in the measurements of the two extremes, we determined that we can reliably measure

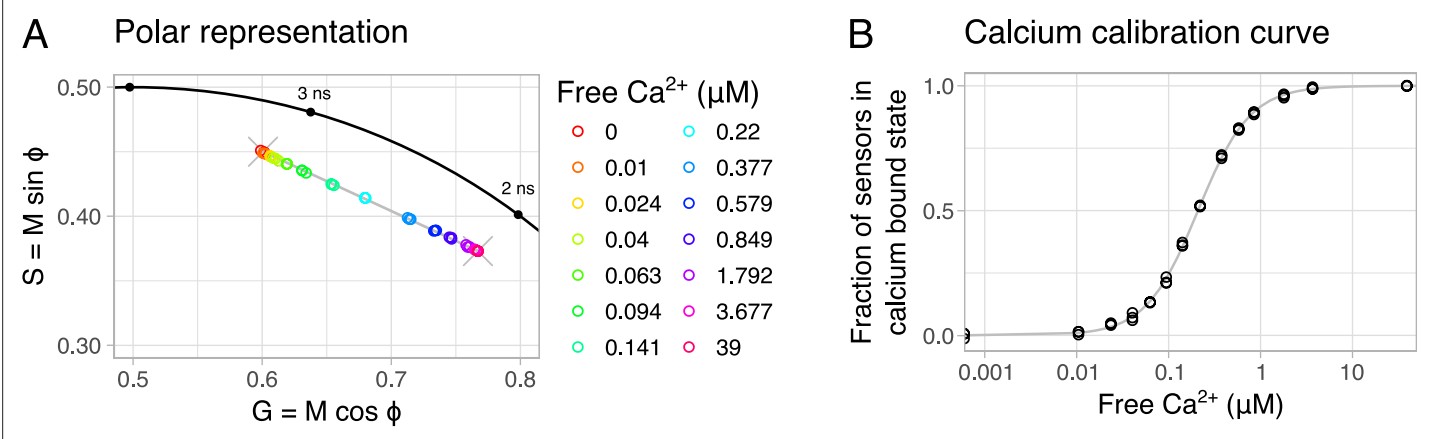

**Figure 3.** Calcium calibration of G-Ca-FLITS in vitro at 37°C. (**A**) The measured fluorescence lifetime of protein isolate of G-Ca-FLITS in a range of calcium concentrations is plotted in a polar space ($n$ = 3), with the color indicating the concentration. The measurements fall on a straight line (in gray) on the polar plot between the lowest and highest concentration (indicated by an X). (**B**) For each measurement, the fraction of sensors in the calcium-bound state (the low lifetime) is determined, taking the intensity contribution of the two extreme states into account. The fraction is plotted against the concentration of free calcium to obtain a calibration curve.

The online version of this article includes the following source data and figure supplement(s) for figure 3:

**Source data 1.** Source data for *Figure 3*.

**Figure supplement 1.** Calcium calibration of G-Ca-FLITS in vitro at 37°C and at RT.

calcium concentration with G-Ca-FLITS at 37°C between 14 nM and 16 µM. The same calibration was also performed at RT (*Table 4*, *Figure 3—figure supplement 1*). Here, the calcium affinity was $K_D$ = 339 nM, higher compared to the calibration at 37 °C. This is in line with the notion that binding strength generally increases with decreasing temperature.

### Little change in intensity between the two states

During engineering and characterization of G-Ca-FLITS, we noted a low intensity contrast for the two extreme states. For a quantitative brightness analysis in cells, we co-expressed G-Ca-FLITS and the red fluorescent protein mScarlet-I in HeLa cells from a single plasmid, using a P2A sequence (*Kim et al., 2011*; *Liu et al., 2017*).

We determined the ratio of green fluorescence over red fluorescence to mitigate the effects of transfection efficiency, timing, and cell thickness. The measurement was performed on both unstimulated cells, with a low intracellular calcium level, and cells stimulated with ionomycin and extra calcium to raise the calcium concentration to saturation of the sensor (*Figure 4*). To normalize the GFP/RFP ratio, we used EGFP-mScarlet-I and a construct without a green FP. G-Ca-FLITS and all its intermediate variants showed a higher intensity in both states compared to mTq2_T203Y. We measured a 0.70-fold change in GFP/RFP ratio for G-Ca-FLITS between the calcium-saturated cells and the unstimulated cells.

**Table 4.** Calcium calibration parameters of G-Ca-FLITS.
The calibration was performed with purified protein at two temperatures. The phase and modulation lifetime of the sensor are indicated for the two states of the sensor. apo – in calcium-free buffer (10 mM EGTA), sat – in buffer with 39 mM free $CaCl_2$, $\tau_\varphi$ and $\tau_M$ – phase lifetime and modulation lifetime, $K_D$ – calcium affinity with 95% confidence interval indicated between curly brackets, $n$ – Hill coefficient with 95% confidence interval indicated between curly brackets.

| Temperature | State | $\tau_\varphi$ (ns) | $\tau_M$ (ns) | $K_D$ {CI} (nM) | $n$ {CI} |
|---|---|---|---|---|---|
| | apo | 2.98 | 3.50 | 209 | 1.53 |
| 37°C | sat | 1.93 | 2.43 | {206–211} | {1.51–1.56} |
| | apo | 3.52 | 3.99 | 339 | 1.67 |
| RT | sat | 2.03 | 2.57 | {335–343} | {1.64–1.70} |

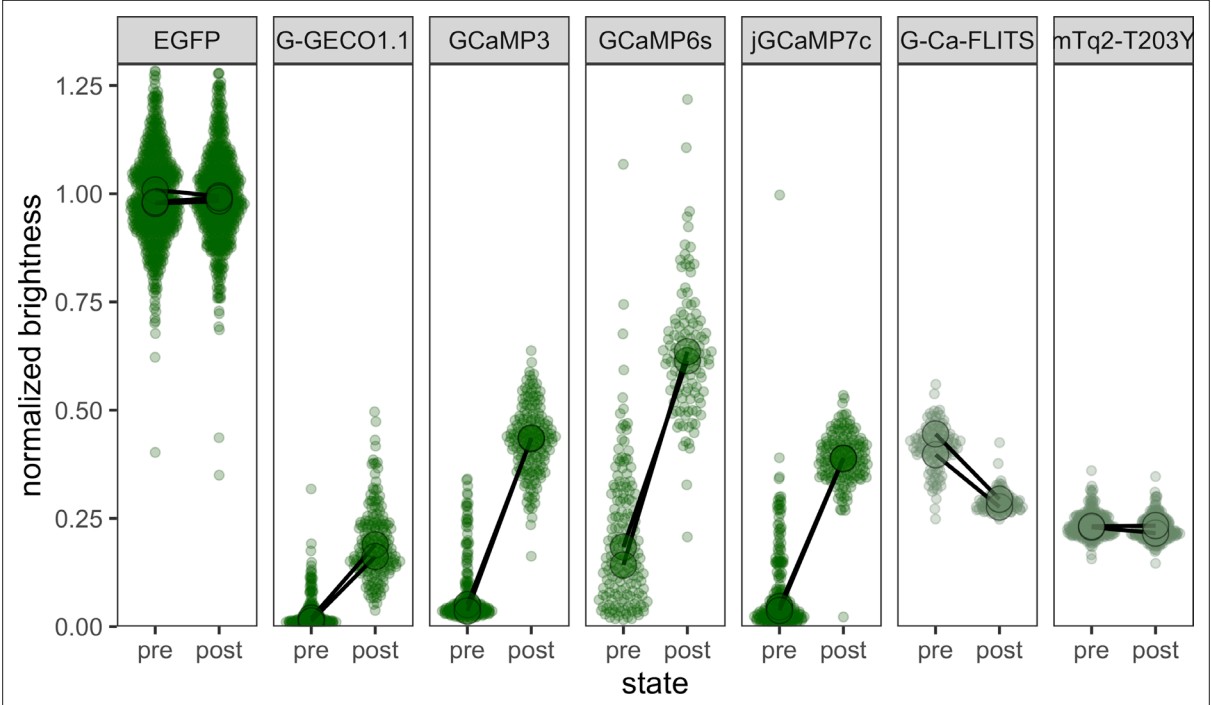

**Figure 4.** Brightness analysis of green fluorescent proteins and biosensors in HeLa cells using a co-expressed mScarlet-I. The ratio GFP/RFP was determined 24 hr after transfection both in resting, unstimulated cells (pre) and cell stimulated with 5 µg/ml ionomycin and 5 mM calcium to reach saturation of the sensors (post). The ratios were normalized to cells expressing EGFP. Dots show individual cell data ($n \geq 99$) and the large dot the average of each biological replicate ($N = 2$, except for EGFP; $N = 3$).

The online version of this article includes the following source data for figure 4:

**Source data 1.** Source data for **Figure 4**.

To compare G-Ca-FLITS to other green GECIs, we repeated the experiment with G-GECO and several variants of GCaMP (**Figure 4**). In contrast to G-Ca-FLITS, these sensors show a high GFP/RFP ratio in the calcium-saturated state and a low ratio in resting cells, resulting in a large ratio difference between the two states. The fold change was 5.6, 3.1, 4.6, and 6.3 for GCaMP3, GCaMP6s, jGCaMP7c, and G-GECO1.1, respectively. The relative intensity of the GCaMPs/GECO in resting cells is much lower compared to the dimmest state of G-Ca-FLITS. We also found a larger standard variation in GFP/RFP in resting cells expressing GCaMP/G-GECO compared to G-Ca-FLITS.

The precision of the fluorescence lifetime depends on the number of photons. We hypothesized that the fluorescence lifetime of G-Ca-FLITS would have a similar precision in both low and high concentrations of calcium, given the small change in intensity. To examine this, we determined the signal to noise ratio (SNR = mean/standard deviation) of the fluorescence lifetime of G-Ca-FLITS in cells, measuring an area of 400 pixels. The absolute SNR is not relevant here, as it will depend on the expression level and acquisition time. But since we have measured the two extremes in the same cells, we can evaluate how the SNR changes between these states for each separate probe (**Figure 5**, **Figure 5—figure supplement 1**). Indeed, the SNR of G-Ca-FLITS is on average only 13% lower for the high calcium state. In comparison, the SNR of Tq-Ca-FLITS increases more than twofold when going from the low to high calcium state, in agreement with its threefold change in intensity. In all cases, the variation of the average lifetime of all cells is similar, meaning the average lifetime in the 400 pixels could be properly determined.

## Measurement of calcium in organelles

Several tags were added to G-Ca-FLITS to target different organelles: the cytosol, the nucleus, the mitochondrial matrix, and the endoplasmic reticulum (ER). With the bright fluorescence of the probe independent of calcium, we could easily confirm the correct localization of the sensor to all targeted organelles, in both HeLa cells and Blood Outgrowth Endothelial Cells (BOECs) (**Figure 6A, B**).

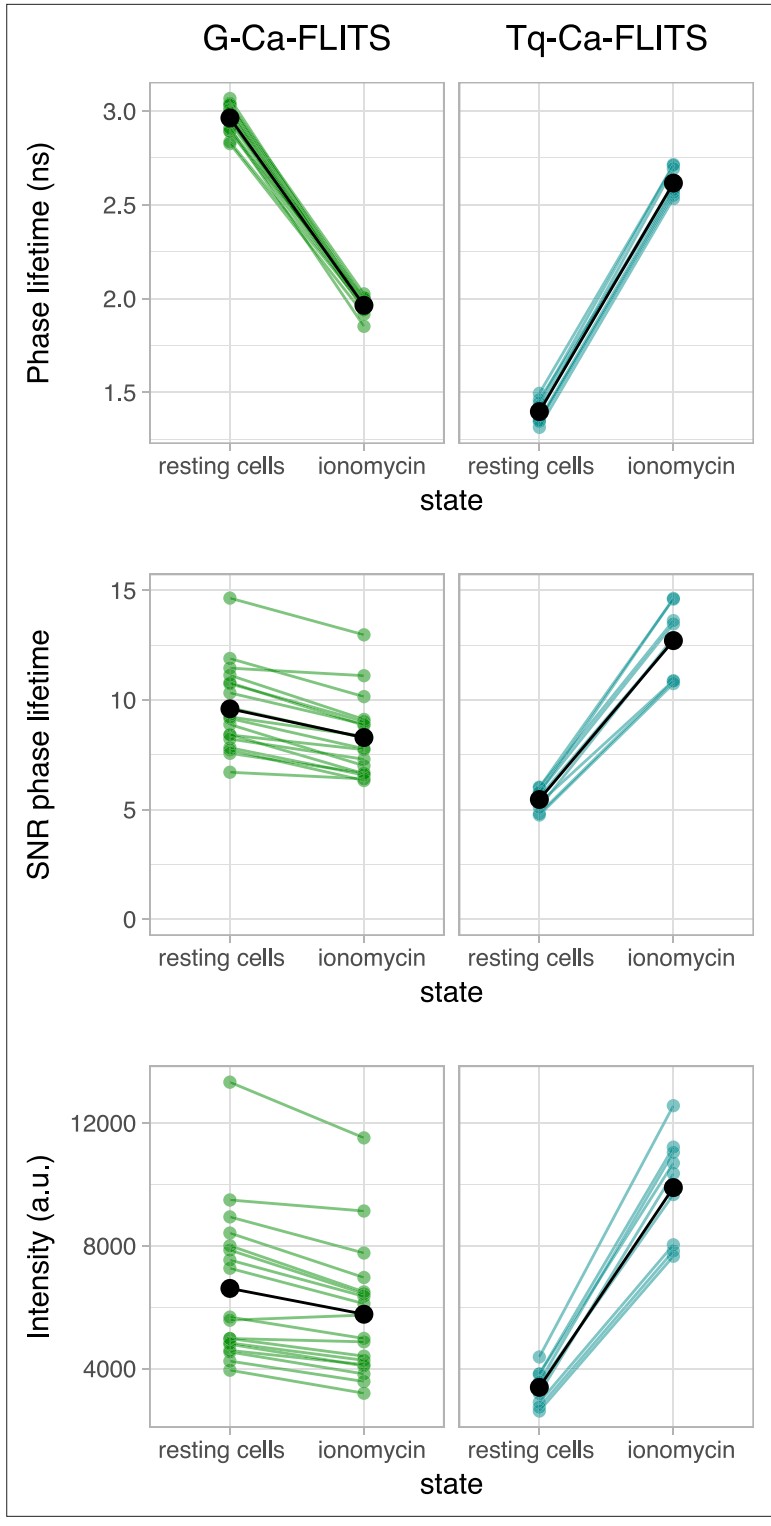

**Figure 5.** Comparison of the signal-to-noise ratio (SNR) of the phase lifetime of G-Ca-FLITS and Tq-Ca-FLITS in HeLa cells. HeLa cells were measured in a resting state and after addition of 5 µg/ml ionomycin and 5 mM calcium. Measurements of individual cells are indicated by colored dots ($n$ = 18 for G-Ca-FLITS, $n$ = 9 for Tq-Ca-FLITS). Lines connect the measurements of the same cell. The average of all cells is shown in black. For each cell, 400 pixels were analyzed for the mean lifetime and intensity, and for the sd of the lifetime. SNR = mean lifetime/sd lifetime. Comparable phase lifetimes are found for all cells expressing a construct, but the SNR is lower for lower intensity cells and varies less between the two states for G-Ca-FLITS.

*Figure 5 continued on next page*

*Figure 5 continued*

The online version of this article includes the following source data and figure supplement(s) for figure 5:

**Source data 1.** Source data for *Figure 5*.

**Figure supplement 1.** Comparison of the signal-to-noise ratio (SNR) of the modulation lifetime of G-Ca-FLITS and Tq-Ca-FLITS in HeLa cells.

Next, we measured the calcium concentration in the various organelles in the same two cell types, using widefield FD-FLIM. Without stimulation, HeLa cells showed an average calcium concentration of 57 nM in the cytosol, 79 nM in the nucleus, and 148 nM in the mitochondria. After the addition of 5 µg/ml ionomycin and 5 mM calcium, the average concentration was raised above 1 µM. In the ER, the concentration was above 1 µM in all experiments. BOECs showed the same pattern, with average concentrations of 73 nM in the cytosol, 101 nM in the nucleus, 119 nM in the mitochondria, and above 1 µM in the ER. After stimulation, the concentration in all organelles was elevated to above 1 µM (*Figure 6C, D*).

In both cell types, no spatial differences were observed in unstimulated cells, except for the mitochondria of HeLa cells. Here, about 20% of the cells had in peripheral parts a higher calcium concentration, comparable to the concentration in mitochondria after stimulation with ionomycin. These areas are not included in the calculations of the aforementioned averages.

HeLa cells expressing G-Ca-FLITS in the cytosol were stimulated with 1 µM histamine. We observed various patterns of oscillating calcium concentrations, confirming that binding of calcium to G-Ca-FLITS is reversible (*Figure 6E*). The calcium concentration was elevated to about 1 µM after addition of ionomycin and calcium, as seen before. The same experiment was repeated with HeLa cells expressing G-Ca-FLITS in the mitochondria (*Figure 6F*). Stimulation with histamine resulted in either no reaction or an elevation of the calcium concentration followed by a drop to basal level. Addition of ionomycin and calcium resulted in high concentrations as expected.

Since confocal imaging of fluorescence lifetime results in images with a better spatial resolution, we used a confocal microscope with a time-correlated single photon counting (TCSPC) detector to examine the heterogeneity that we observed in mitochondria. From the confocal lifetime images, it is evident that several cells have mitochondria that show a decreased lifetime in the periphery of the cell (*Figure 6—figure supplement 1*). This indicates that mitochondria show heterogeneity and that a pool of mitochondria has increased calcium levels.

The confocal TCSPC system needs relatively long integration times (several minutes), and we turned to a Leica Stellaris system which allows for a >10-fold higher temporal resolution to study calcium dynamics. We introduced the 4mts-G-Ca-FLITS into HeLa cells and acquired a timelapse of lifetime images. At the end of the experiment, digitonin was added to acquire the minimal lifetime, reflecting maximal calcium. Using the calibration curve that was generated using purified protein incubated with different calcium buffers (*Figure 7—figure supplement 1*), we transformed the FLIM data into calcium concentration images as we did before. The calcium concentration images show the heterogeneity in calcium concentration and its dynamics over time (*Figure 7*). The elevated calcium concentrations are reversible but can last several minutes. Intriguingly, there is a substantial difference in the dynamics between patches of mitochondria.

## Multiplex imaging

The green FLITS probe has a red shift of approximately 35 nm in both excitation and emission peaks relative to its Turquoise parent, resulting in a strikingly different emission color (*Figure 1*). Therefore, we examined whether we can use these probes together in a single cell for multiplex imaging. To this end, we co-expressed a nuclear targeted 3xnls-Tq-Ca-FLITS and a G-Ca-FLITS-3xnes with a triple nuclear export sequence (nes). We selected an excitation wavelength and an emission bandpass to selectively image either of the two probes and, as a result, we could distinguish the differential localization of the two probes (*Figure 8*). Upon stimulation of the cells with histamine, we observed a lifetime change for both probes, with the lifetime of Tq-Ca-FLITS increasing and the lifetime of G-Ca-FLITS decreasing as expected, since histamine increases the intracellular calcium concentrations.

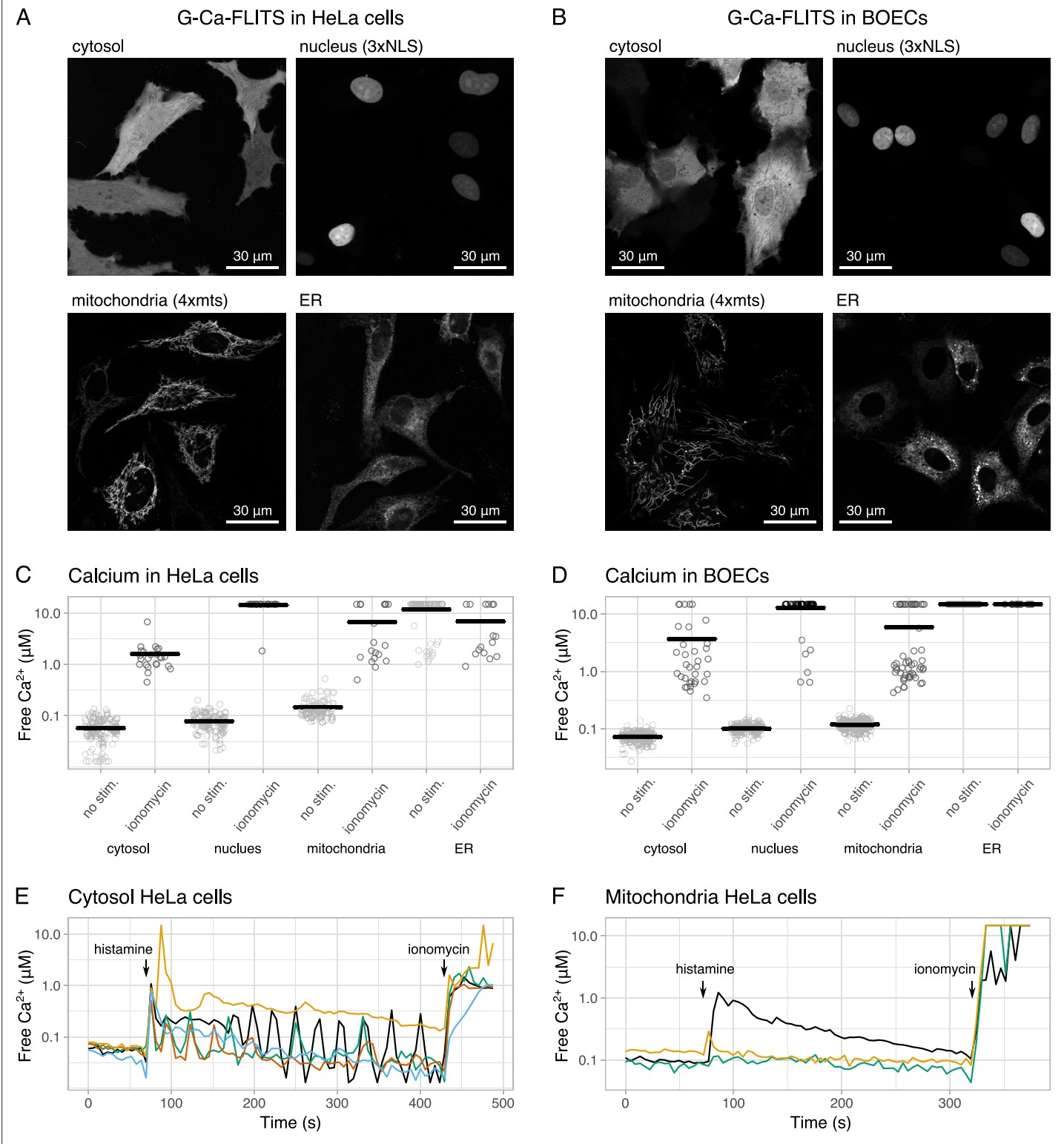

**Figure 6.** Measuring calcium concentrations in HeLa cells and Blood Outgrowth Endothelial Cells (BOECs) using G-Ca-FLITS targeted to various organelles. (**A**) Localization of G-Ca-FLITS in HeLa cells and (**B**) in BOECs. Images are taken with a ×63 magnification. (**C**) Measured calcium concentration in various organelles in HeLa cells and (**D**) BOECs. Measurements of single cells are indicated by circles, gray for unstimulated cells and black for cells after addition of 5 µg/ml ionomycin and 5 mM calcium. The mean of all cells is indicated by a black line. (**E**) Changing free calcium concentration after stimulation with 1 µM histamine or addition of 5 µg/ml ionomycin and 5 mM calcium in the cytosol and (**F**) in the mitochondria. Each line represents a single cell. Arrows indicate the moments of addition.

*Figure 6 continued on next page*

*Figure 6 continued*

The online version of this article includes the following source data and figure supplement(s) for figure 6:

**Source data 1.** Source data for *Figure 6C-F*.

**Figure supplement 1.** Confocal lifetime imaging of G-Ca-FLITS expressed in mitochondria of HeLa cells.

**Figure supplement 2.** Aggregated FLIM data of measurements in organelles under resting conditions and after incubation with ionomycin.

## 2P-FLIM

To assess the functionality of G-Ca-FLITS and Tq-Ca-FLITS with two-photon TCSPC FLIM microscopy, we measured calcium in the brains of *Drosophila melanogaster* individuals affixed to a previously described stage (*Maimon et al., 2010*). Studies of activity-dependent transcription in neurons have used saline with elevated $[K^+]$ to induce sustained and controlled increases in intracellular calcium. We employed a similar approach to raise external $[K^+]$ from 3 mM to 103 mM by perfusing saline over the brain while imaging jGCaMP7f (*Dana et al., 2019*), Tq-Ca-FLITS, and G-Ca-FLITS signals. Our experiments here focus on the EPG neurons, a population of neurons innervating the ellipsoid body region of the fly brain and which serve a role in navigation-related computations (*Green et al., 2019*; *Seelig and Jayaraman, 2015*; *Figure 9A*).

jGCaMP7f exhibited large changes in fluorescence intensity several minutes after we elevated external $[K^+]$ (*Figure 9B*). After a number of minutes (with the exact number varying from fly to fly), calcium levels fluctuated between low and high states synchronously across the field of view (including the EPG neurites visible in the middle of *Figure 9E* and the neurites of off-target cells in the bottom corners of the same images). The sample with jGCaMP7f showed only a very small change in lifetime in response to elevated calcium (0.030 ns ± 0.003, mean ± sd) despite the many-fold shift in fluorescence intensity 5.79 $\Delta F/F$ ± 0.503 (mean ± sd).

Tq-Ca-FLITS fluorescence was higher than that of jGCaMP7f at baseline (*Figure 9E*). In contrast, despite the high brightness of G-Ca-FLITS in the apo state in other systems, the baseline intensity of G-Ca-FLITS in these fly neurons was much lower than both jGCaMP7f and Tq-Ca-FLITS, for unclear reasons. Even with increased laser power, G-Ca-FLITS was dimmer than jGCaMP7f at baseline (see Methods) (*Figure 9E*, top row).

We evaluated the relative brightness of purified Tq-Ca-FLITS and G-Ca-FLITS on beads by either 1-Photon Excitation (1PE) (at 460 nm) or 2-Photon Excitation (2PE) (at 920 nm) and observed a similar brightness between the two modes of excitations (*Figure 9—figure supplement 1*). This shows that

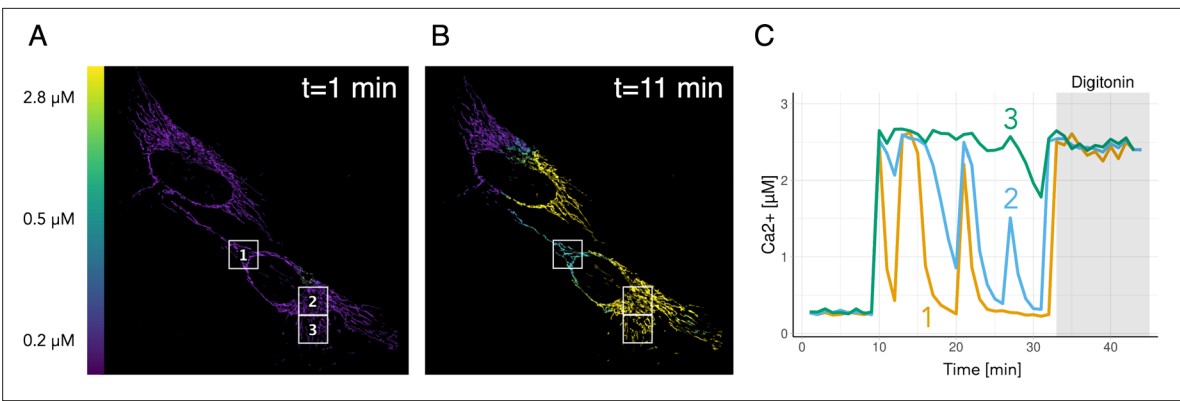

**Figure 7.** Quantitative imaging of spontaneous calcium dynamics in mitochondria. (**A, B**) FLIM images were acquired with a Leica Stellaris 8 every 60 s and at the end of the sequence 10 μM digitonin was added to obtain a maximal response. The lifetime images were converted to calcium concentration using the $K_D$ determined in vitro and the extremes from the lifetime image. False colors reflect the calcium concentration, according to the scale bar on the left. The size of the images is 132 μm x 132 μm. Calcium image at $t = 1$ (**A**) and $t = 11$ min (**B**), showing the three regions where calcium concentrations were quantified and displayed over time (**C**).

The online version of this article includes the following source data and figure supplement(s) for figure 7:

**Source data 1.** Source data for *Figure 7C*.

**Figure supplement 1.** In vitro calibration of G-Ca-FLITS on the Leica Stellaris 8 at room temperature.

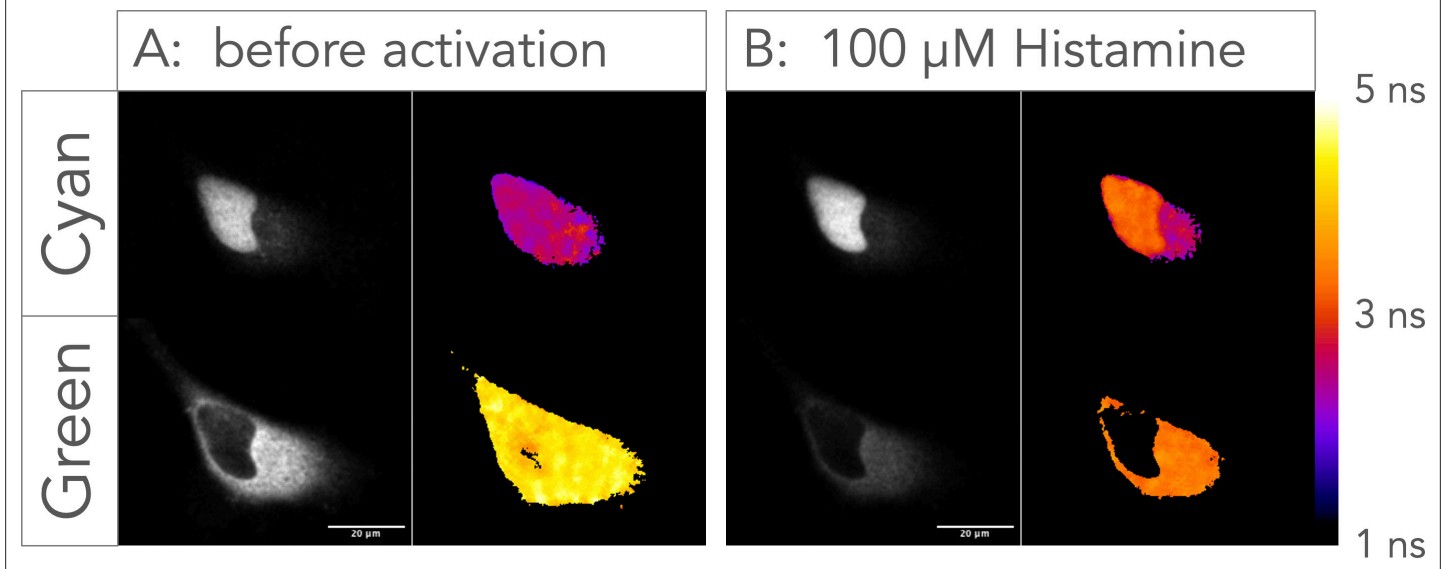

**Figure 8.** Multiplex imaging of Tq-Ca-FLITS and G-Ca-FLITS. FLIM images of HeLa cells co-transfected with 3xnls-Tq-Ca-FLITS and G-Ca-FLITS-3xnes were acquired with a Leica Stellaris 8 before (**A**) and after addition of 100 µM histamine (**B**). False colors reflect the lifetime (average arrival time), according to the scale bar on the right. For the cyan channel, the excitation was at 440 nm and detection at 450–490nm and for the green channel, the excitation was at 490 nm and detection at 500–540nm.

the two probes have similar efficiencies in 2PE and suggests that the low brightness of G-Ca-FLITS in *Drosophila* is due to lower expression or poor folding.

In response to elevated [K⁺], Tq-Ca-FLITS showed large increases in fluorescence (1.672 $\Delta F/F$ ± 0.299, mean ± sd) and empirical lifetime (0.893 ns ± 0.043, mean ± sd) (***Figure 9F, G***). The response of G-Ca-FLITS was also consistent with increases in intracellular calcium: this sensor exhibited large decreases in empirical lifetime (–0.465 ns ± 0.101 (mean ± sd)), and a modest decrease in intensity (–0.339 $\Delta F/F$ ± 0.043, mean ± sd) after perfusion of elevated external [K⁺] (***Figure 9F–H***).

Overall, all three fluorophores exhibit calcium-dependent signals in an intact brain using 2P-FLIM. The Tq-Ca-FLITS lifetime response is particularly large and shows much higher brightness under 2P excitation than G-Ca-FLITS in both the baseline and high calcium states.

## Discussion

Most fluorescent biosensors toggle between a dim and a bright state. The dim fluorescence decreases the SNR, thereby increasing the measurement error in quantitative imaging. Here, we present a green calcium biosensor, G-Ca-FLITS, that maintains a consistent brightness over its full dynamic range. The calcium-dependent lifetime decrease of 1.2 ns when calcium is elevated can be used to quantify calcium concentrations in cells. In comparison, the cyan/turquoise sensor Tq-Ca-FLITS we previously developed (***van der Linden et al., 2021***) shows a threefold change in intensity combined with a similar change in lifetime. The intensity of G-Ca-FLITS is higher than the dim (calcium-free) state of several popular green GCaMPs and comparable to the bright state of most tested sensors, as determined by ratio imaging with co-expression of mScarlet-I. Of the tested GCaMPs, GCaMP6s seems to have a fairly bright calcium-free state, and we believe this is due to the higher calcium affinity ($K_D$ = 144 nM) of this sensor (***Chen et al., 2013***). In contrast, the affinities are 339, 298, and 618 nM at room temperature, for G-Ca-FLITS, jGCaMP7c (***Dana et al., 2019***) and G-GECO1.1 (***Zhao et al., 2011***), respectively.

The minimal change in fluorescence intensity compared to other commonly used calcium biosensors results in a similar SNR for both states of G-Ca-FLITS. Another advantage of a consistently bright probe is that cells and structures labeled with the biosensor can be readily visualized. Therefore, low concentrations of calcium can be precisely measured in unstimulated mammalian cells, as well as increased concentration after stimulation. The quantitative measurements require calibration of the biosensor response to different calcium concentrations. Although the calibration is equipment independent under ideal conditions and theoretically only needs to be performed once, we do see

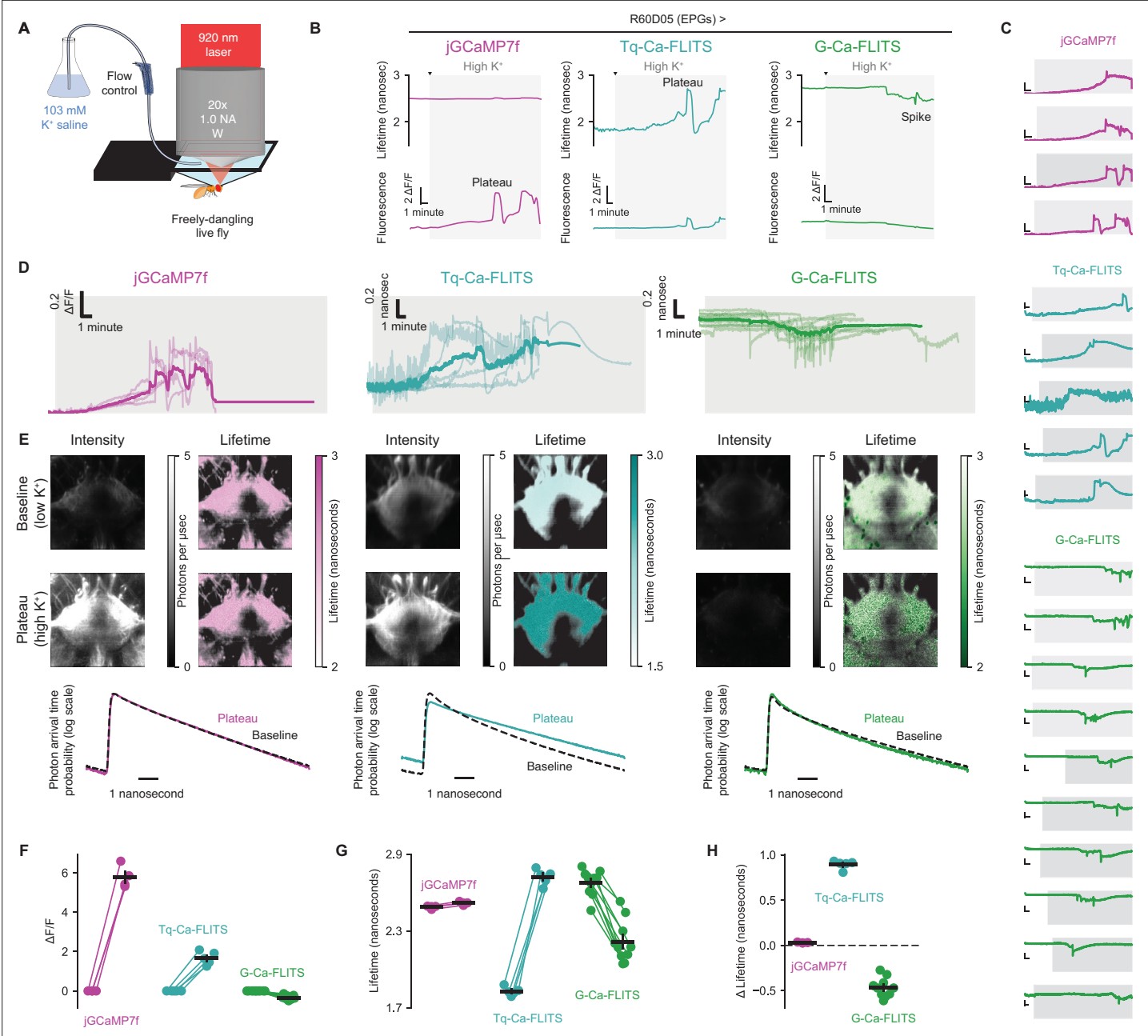

**Figure 9.** Two-photon time-correlated single photon counting (TCSPC) signals from G-Ca-FLITS and Tq-Ca-FLITS, and comparison to jGCAMP7f, in an intact *Drosophila* brain. (**A**) Schematic of a head-fixed live *Drosophila melanogaster* female, imaged with standard saline for 2 min before perfusing a high [K+] saline variant over the brain. (**B**) Single example traces of flies expressing jGCaMP7f (pink, left), Tq-Ca-FLITS (turquoise, center), or G-Ca-FLITS (green, right) in the EPG neurons under the control of R60D05-Gal4. Top row: Tq-Ca-FLITS and G-Ca-FLITS show strong FLIM changes in response to elevated [K+] while jGCaMP7f shows very little. Bottom row: jGCaMP7f and Tq-Ca-FLITS exhibit large changes in fluorescence while G-Ca-FLITS becomes moderately dimmer. Gray region indicates time when high K+ saline was perfused. (**C**) Intensity traces from all flies. Vertical scalebars: 0.200 ns for G-Ca-FLITS and Tq-Ca-FLITS, 5 ΔF/F for jGCaMP7f. Horizontal scalebars: 1 min. Gray region indicates time when high K+ saline was perfused. (**D**) Aggregated responses, the thick lines represent the mean response of all measurements. (**E**) Fluorescence (left) and FLIM (right) images (72 x 72 μm) of example flies from panel B before [K+] elevation (baseline, 2 min average, top row) and for the 30 s surrounding the peak change in calcium (second row). Lifetime images are masked to show only automatically determined 'foreground' pixels. Third row: photon arrival time histograms for the data of the above rows. (**F–H**) Summary statistics for all flies (from panel **C**). (**F**) shows changes in fluorescence intensity, (**G**) shows absolute lifetime measurements, and (**H**) reflects the change in lifetime from baseline to plateau for each of the three indicators.

The online version of this article includes the following source data and figure supplement(s) for figure 9:

**Source data 1.** Source data for *Figure 9B-D*.

*Figure 9 continued on next page*

*Figure 9 continued*

**Source data 2.** Source data for *Figure 9F-H*.

**Figure supplement 1.** Similar brightness for G-Ca-FLITS and Tq-Ca-FLITS between 1-Photon Excitation (1PE) and 2-Photon Excitation (2PE).

differences between different setups. Therefore, it is necessary to repeat the calibration for different setups. We prefer to use the phasor analysis (which can be used for both frequency- and time-domain FLIM), as it makes no assumptions about the underlying decay kinetics.

Our measured concentrations in resting cells agree with literature, 20–100 nM is reported for cytoplasm and nucleus, and a slightly higher concentration of 50–300 nM in the mitochondria (*Arnaudeau et al., 2001*; *Berridge et al., 2000*; *Dalal et al., 2020*; *Palmer et al., 2006*; *Rotrosen and Gallin, 1986*; *Worthen and Nollert, 2000*).

We note that the underlying assumption of the quantification of organellar calcium concentrations is that the lifetime contrast is the same in each organelle. This is broadly true for most of the measurements (*Figure 6—figure supplement 2*). Yet, there are also differences. It is currently unclear whether the discrepancies are due to differences in the physicochemical properties of the compartments, or whether there is a technical reason (the efficiency of ionomycin for saturating the biosensor in the different compartments is unknown, as far as we know). This is something that is worth revisiting. A related issue that deserves attention is the level of agreement between in vitro and in vivo calibrations.

All calcium indicators also act as buffers, and this limits the accuracy of the absolute measurements, especially for the lower calcium concentrations (*Rose et al., 2014*), as the expression of the biosensor is usually in the low micromolar range. Free calcium in the ER is reported to be >300 µM, well above saturation of G-Ca-FLITS (*Arnaudeau et al., 2001*).

Using G-Ca-FLITS, we were able to detect various (oscillating) patterns of calcium in HeLa cells after stimulation with histamine. These data show that the sensor is suitable for detection of repeated transient calcium elevations. Importantly, G-Ca-FLITS gives a stable read-out at the high pH of 8.0 in mitochondria. We observed high calcium levels in some peripheral mitochondria in HeLa cells, but not in endothelial cells. Others have shown heterogeneous responses of calcium (*Greotti et al., 2019*; *Palmer et al., 2006*) and electrical signals (*Collins et al., 2002*) in mitochondria. However, this is different from the sustained high levels we observed in exclusively peripheral parts of the cells, without any stimulation. Studies show that mitochondria can be distinguished in different pools based on their location and function (*Ngo et al., 2021*; *Park et al., 2001*). Peripheral mitochondria are one of these groups and have less connections to the ER compared to other pools. These connections play an important role in calcium signaling of the mitochondria (*Giorgi et al., 2018*). In cancerous cells, like HeLa cells, the mitochondrial calcium homeostasis can be dysregulated (*Delierneux et al., 2020*). We speculate that this leads to the sustained high levels of calcium in some peripheral mitochondria, possibly linked to aberrant connections to the ER.

To our knowledge, the sustained high levels we observed in mitochondria have not been reported yet. The types of probes used so far might be a contributing factor. Detecting the sustained high levels without stimulation is hardly possible with a probe that only shows an intensity response, like the widely used Rhod-2 or Aequorin. It is possible by using filter Förster Resonance Energy Transfer (FRET)/sensitized emission (*Gadella, 2009*), but this requires extensive controls. Also, much research in mitochondria has been focused on the rise in calcium after stimulation, and therefore probes with a high $K_D$ (over 1 µM) have been used preferably (*Arnaudeau et al., 2001*; *Fernandez-Sanz et al., 2019*; *Fonteriz et al., 2010*; *Palmer et al., 2006*). This low affinity might be too low to detect the effect we have seen.

G-Ca-FLITS shares its >1 ns lifetime contrast and pH stability with the cyan/turquoise sensor Tq-Ca-FLITS we previously developed (*van der Linden et al., 2021*), but the green emission is advantageous for several reasons: (1) excitation for green FPs is installed on virtually every lifetime microscope, while excitation for cyan FPs is less common, (2) the phototoxicity will be less and the penetration depth will be improved, and (3) it provides possibilities for multiplexing with FRET-FLIM sensors that often harbor a cyan-yellow FP pair, for example EPAC (*Klarenbeek et al., 2015*).

Since biosensors with a CFP-like chromophore are largely pH insensitive (*van der Linden et al., 2021*; *Zhong et al., 2024*), and we show here that the pH independence is retained for the Green Ca-FLITS, we expect that adding the T203Y mutation to a cyan sensor is a good approach for generating pH-insensitive green lifetime-based sensors.

The unique properties of the new green calcium sensor are evident from its photophysical properties. The extinction coefficient of the calcium-bound and -free states of G-Ca-FLITS is very similar, but the difference in QY is significant. Interestingly, the lower QY is observed in the calcium-bound state and is very similar to the QY of mTq2_T203Y, the FP that is used in this probe. This, combined with the resemblance in spectral shape and pH sensitivity, indicates that in the calcium-bound state, the FP-barrel is probably in roughly the same closed conformation as mTq2_T203Y. When calcium is removed, a more open state is assumed with a more favorable QY and a higher lifetime. Possibly, the lifetime contrast could be further improved by increasing the QY of the calcium-free state.

Usually, a higher QY results in a higher intensity; however, in G-Ca-FLITS, the open state has a differential shaped excitation spectrum which leads to a decreased excitation efficiency. These effects combined have resulted in a sensor where the two different states have a similar intensity despite displaying a large lifetime contrast. We note that this effect is dependent on the excitation wavelength. The smallest intensity change is obtained when the calcium-bound state is preferably excited (i.e. near 450 nm), and the effect is less pronounced when the probe is excited near its peak at 474 nm.

We evaluated the use of Tq-Ca-FLITS and G-Ca-FLITS for 2P-FLIM and observed a surprisingly low brightness of the green variant in an intact fly brain. This result is consistent with a study finding that red-shifted fluorescent-protein variants that are much brighter under one-photon excitation are, surprisingly, dimmer than their blue cousins in multi-photon microscopy (*Molina et al., 2017*). The responses of both probes were in line with their properties in single photon FLIM, but given the low brightness of G-Ca-FLITS under 2PE in *Drosophila*, the Tq-Ca-FLITS is a better choice in this system. Yet, the brightness of G-Ca-FLITS with 2PE at 920 nm is comparable to Tq-Ca-FLITS, so we expect that 2P-FLIM with G-Ca-FLITS is possible in tissues that express it well.

In conclusion, we developed a green fluorescent GECI, named G-Ca-FLITS, with a lifetime contrast of over 1 ns, and a modest intensity change. This sensor does not have one (very) dim state, unlike other green GECIs; therefore, the variation in the measurement is comparable in both states. In addition, the lifetime response is stable above pH 7. We successfully measured the calcium concentration in various organelles in both HeLa cells and BOECs, including the mitochondria. A portion of HeLa cells shows sustained high levels of calcium in some peripheral mitochondria, but further research is required to determine the origin of this heterogeneity. We anticipate that the G-Ca-FLITS will be a valuable tool to examine the heterogeneity of calcium concentrations in mitochondria and other cellular compartments.

## Materials and methods

**Key resources table**

| Reagent type (species) or resource | Designation | Source or reference | Identifiers | Additional information |
|---|---|---|---|---|
| Strain, strain background (*Escherichia coli*) | E.cloni 10G | Lucigen | 60106-1 | Electrocompetent cells |
| Genetic reagent (*D. melanogaster*) | UAS-Tq-Ca-FLITS | This paper | | y[1]w[1118]; PBac{y[+mDint2]w[+mC]=20XUAS-IVS-Tq-Ca-FLITS}VK00027 |
| Genetic reagent (*D. melanogaster*) | UAS-G-Ca-FLITS | This paper | | y[1]w[1118]; PBac{y[+mDint2]w[+mC]=20XUAS-IVS-G-Ca-FLITS}VK00027 |
| Cell line (*Homo sapiens*) | HeLa | ATCC | CCL-2 | |
| Cell line (*Homo sapiens*) | cbBOEC | Jaap van Buul | | |
| Recombinant DNA reagent | G-Ca-FLITS (plasmid) | This paper | https://www.addgene.org/191465/ | Cytoplasm targeted biosensor |
| Recombinant DNA reagent | 4xmts-G-Ca-FLITS (plasmid) | This paper | https://www.addgene.org/191462/ | Mitochondria targeted biosensor |
| Recombinant DNA reagent | ER-G-Ca-FLITS-KDEL (plasmid) | This paper | https://www.addgene.org/191464/ | ER targeted biosensor |

*Continued on next page*

*Continued*

| Reagent type (species) or resource | Designation | Source or reference | Identifiers | Additional information |
|---|---|---|---|---|
| Recombinant DNA reagent | 3xnls-G-Ca-FLITS (plasmid) | This paper | https://www.addgene.org/191463/ | Nucleus targeted biosensor |
| Chemical compound, drug | Polyethylenimine, Linear, MW 25000, Transfection Grade | Polysciences | 23966 | |
| Chemical compound, drug | Ionomycin | L C Laboratories | I-6800 | 5 µg/ml |
| Commercial assay, kit | Calcium Calibration Buffer Kit #1, zero and 10 mM CaEGTA | Thermo Fisher Scientific | C3008MP | |
| Software, algorithm | R | RStudio | | |
| Software, algorithm | FIJI | doi:https://doi.org/10.1038/nmeth.2019 | https://github.com/fiji/fiji | |
| Software, algorithm | FIJI | doi:https://doi.org/10.1038/nmeth.2019 | https://github.com/fiji/fiji; *Schindelin et al., 2025* | |

## General cloning

We used *E. coli* strain *E. cloni* 5-alpha (short: *E. cloni*, Lucigen corporation) for all cloning procedures. For DNA assembly, competent *E. cloni* was transformed using a heat shock protocol according to the manufacturers' instructions. For protein expression, *E. cloni* was grown using super optimal broth (SOB, 0.5% (wt/vol) yeast extract, 2% (wt/vol) tryptone, 10 mM NaCl, 20 mM MgSO$_4$, 2.5 mM KCl) supplemented with 100 µg/ml kanamycin and 0.2% (wt/vol) rhamnose (SKR). For agar plates, 1.5% (wt/vol) agar was added. Bacteria were grown overnight at 37°C. Plasmid DNA was extracted from bacteria using the GeneJET Plasmid Miniprep Kit (Thermo Fisher Scientific) and the obtained concentration was determined by Nanodrop (Life Technologies).

DNA fragments were generated by PCR, using Pfu DNA polymerase (Agilent Technologies) unless otherwise indicated. DNA fragments were visualized by gel electrophoresis on a 1% agarose gel, run for 30 min at 80 V. PCR fragments were purified using the GeneJET PCR purification Kit (Thermo Fisher Scientific) and digested with restriction enzymes to generate sticky ends. Restriction enzymes were heat inactivated at 80°C for 20 min if necessary. Vector fragments were generated by restriction of plasmids and the correct bands were extracted from gel using the GeneJET Gel Extraction Kit (Thermo Fisher Scientific). DNA fragments were ligated using T4 DNA ligase (Thermo Fisher Scientific), per the manufacturers' protocol.

For targeted mutagenesis, we used the protocol as described before (*Bindels et al., 2014*). Primers were designed with an annealing region of 15 bp both left and right of the desired mutation(s). The full plasmid was amplified by PCR, and the DNA was digested with DpnI to remove template DNA. *E. cloni* was transformed with the digested PCR mix, and individual colonies were selected for isolation of individual colonies.

Correct construction of plasmids was verified by control digestion and sequencing (Macrogen Europe). All primers (*Supplementary file 2*) were ordered from Integrated DNA Technologies.

## Engineering of G-Ca-FLITS

Tq-Ca-FLITS was subjected to mutagenesis of residues T203 and I167 (both mq2 numbering), by cloning using in vitro assembly (IVA) (*García-Nafría et al., 2016*). First, full plasmids were replicated by PCR while introducing the desired mutation using the dual expression plasmid pFHL-Tq-Ca-FLITS (Addgene plasmid #129628) as template, primers 1–2, 3–4, or 5–6 and Phusion DNA polymerase (Thermo Fisher Scientific). The resulting PCR mix was DpnI digested, and 4 µl of the mix was used for transformation of *E. cloni*. The linear DNA fragments were assembled into a plasmid in bacteria using a 15-bp homologous region at the ends of the fragments. In addition, intermediate, differently

circular permutated variants of Tq-Ca-FLITS were subjected to mutagenesis of T203 by using primers 1–2 and the IVA method described above. Individual colonies were picked for testing of the excitation and emission spectra, for testing of the lifetime response and for plasmid isolation and sequencing.

The best-performing variant was subjected to further mutagenesis of the sequences linking the FP with the CaM and M13 peptide. A library of variants was created by PCR with pFHL-Tq-Ca-FLITS as template and primers 7–8. The PCR fragments and a pFHL backbone (containing R-GECO1, a red calcium sensor) were digested using SacI and MluI, followed by ligation. Individual colonies were picked for screening of the excitation and emission spectra and of the lifetime response. Plasmids of interesting variants were isolated for sequencing.

## Generation of the dual expression vector for ratiometric measurements

The SacI restriction site was removed from mScarlet-I in an 'empty' pDress plasmid (*Bindels et al., 2020*), containing mScarlet-I linked to an anti-FRET linker and a P2A site (Addgene plasmid #130509, but with the mScarlet protein between the AgeI and BsrGI sites replaced with the DNA sequence GCCTCAACATGA and mTurquoise2 replaced by mScarlet-I). The SacI site was removed by directed mutagenesis using primers 9–10, see '**General cloning**'. After mutagenesis, correct removal of the SacI site and the absence of unwanted spontaneous mutations was verified by control digestion with SacI and NheI followed by sequencing. This yielded pDress_mScI_antiFRET_P2A_linker-noSacI (internal number 5517).

An insert fragment was amplified from the new plasmid by PCR using primers 11–12, containing the last part of the rhamnose promoter, the ribosome-binding site, the adapted kozak sequence, the 6xHis-tag, the thrombin recognition site, the mScarlet-I fluorescent protein, the anti-FRET linker, and the P2A sequence. A PCR using Phusion polymerase (Thermo Fisher Scientific) was performed on dual expression plasmid pFHL (Addgene plasmid #129628) using primers 13–14 to generate a backbone fragment containing the Tq-Ca-FLITS calcium sensor, the backbone of the pFHL plasmid, the CMV promoter, and the first part of the rhamnose promoter. Both the insert and backbone fragments were digested with BamHI and EcoRI to create overhanging ends. The fragments were ligated together and *E. cloni* was transformed with the DNA. We named this plasmid pFR_mScarletI-antiFRET-P2A-mTurquoise2_T203Y. Correct construction of the new pFR (Franka Ratio) plasmid was verified by control digestion with NheI and BsrGI and sequencing.

G-Ca-FLITS and intermediate variants were put in the ratioplasmid pFR by restriction (MluI and SacI) and ligation of pFR_mScarletI-antiFRET-P2A-mTurquoise2_T203Y and a pFHL plasmid containing the sensor of interest. Correct insertion was verified by sequencing.

As a control, also EGFP and green variants of mTq2 and sfmTq2 were put in the pFR plasmid. The FPs mTq2 and sfmTq2 were subjected to directed mutagenesis to create mTq2_T203Y and sfmTq2_T203Y with primers 15–16. Next, PCR fragments were created of the green turquoise FPs and EGFP by using primers 17–18. PCR fragments and a pFR backbone were digested with BamHI and HindIII, followed by ligation. Correct insertion was verified by sequencing.

Green GECIs were put in pFR for comparison. DNA fragments were created by PCR using pGD-CMV-jGCaMP7c (Addgene plasmid #105320), pcDNA3-Cyto-GCaMP3 (Addgene plasmid #64853), pGP-CMV-GCaMP6s (Addgene plasmid #40753) and CMV-G-GECO1.1 (Addgene plasmid #32445) as template and primers 19–20. The fragments and a pFR backbone were digested with BamHI and HindIII and ligated together. Correct insertion was verified by sequencing.

## Plasmids for expression in organelles

The sensor sequence was amplified by PCR to generate cytoplasmic, nuclear, mitochondrial, and ER-targeted versions of G-Ca-FLITS for transient transfection of mammalian cells, using primers 21–22 (for ER) or 21 and 23 (all others). DNA fragments and vectors carrying the desired tag were digested with AgeI and BsrGI, followed by ligation. Of the following vectors, the FP was exchanged for Tq-Ca-FLITS: 3xnls-mTurquoise2 (Addgene plasmid #98817) for a nuclear tag, 4xmts-mScarlet-I (Addgene plasmid #98818), ER-mScarletI (Addgene plasmid #137805), and mVenus-N1 (color variant of Addgene plasmid #54843) for an untagged version. The G-Ca-FLITS-3xnes (3x nuclear export sequence) was made by ligating it into 3xNLS-SGFP2(T65G)-T2A-SGFP2-3xNES (Addgene plasmid # 60492) that was cut with NdeI/BsrGI.

## Screening of sensors in bacterial lysate

*E. cloni* bacteria were used for screening of green calcium sensor variants. Bacteria expressing (a library of) sensor variants on pFHL plasmids were grown on SKR-agar plates. Colonies visually showing a (bright) green fluorescence (excitation filter 470/40 nm, emission filter 525/50 nm) under a stereo-microscope (MZFLIII, Leica) were selected. Bacteria were grown overnight in 1.5 ml SKR in a deep well plate (732–2893, VWR) covered with an adhesive seal (AB-0718, Thermo Fisher Scientific) at 37°C while shaking at 280 rpm. The next day, 1 µl of culture was spotted on a SKR plate for storage, and the remainder of the bacteria was harvested by 15 min centrifugation at 2683 × $g$ in a swing-out centrifuge (5810 R, Eppendorf). Supernatant was removed and 200 µl of lysis buffer (2% Deoxycholic acid in 50 mM Tris-HCl pH 8.0) was added. During optimization of the lysis protocol, the lysis buffer consisted of 50 mM Tris-HCl pH 8.0 with DOC (0, 1, 2, or 3%), urea (0, 0.5, or 1 M), lysozyme (1 mg/ml), sodium dodecyl sulfate (0, 0.1, 0.25, or 0.5%) and/or Triton X-100 (0, 0.1, 0.5, or 1%). Bacteria were brought in suspension using a vortex and incubated for 15 min at RT while intermittently shaking. If indicated, the plate with suspensions was frozen at –20°C and thawed. Cell debris was removed by 40 min centrifugation at 2683 × $g$ in a swing-out centrifuge. The bacterial lysate was collected and stored overnight at 4°C. The next day, bacterial lysates were diluted 2.5–4 ×, in 50 mM Tris-HCl pH 8.0 to a final volume of 200 µl in 96-well plates with black walls and a glass bottom (89626, Ibidi).

A JASCO FP-8500 spectrofluorometer with microplate reader adapter was used to measure the excitation and emission spectra. Excitation spectra were measured at 370–490 nm with a 2.5-nm bandwidth and 1-nm intervals, while collecting emission at 530 nm with 10 nm bandwidth, using a scan speed of 1000 nm/min, 50 ms response time, and a high sensitivity. Emission spectra were measured at 460–600 nm with a 2.5-nm bandwidth and 1-nm intervals, while exciting at 425 nm with 5 nm bandwidth, using a scan speed of 1000 nm/min, 0.1 s response time, and a high sensitivity. Each spectrum was corrected for background by subtraction of the spectrum of a well with only buffer. The same plate was used for determination of the fluorescence lifetime of the lysates as described under '**Lifetime imaging**'. Spectra and lifetimes were recorded after addition of 0.1 mM $CaCl_2$ (10 µl of 2 mM) to saturate the sensors, and again in the same wells after addition of 9.5 mM EDTA (10 µl of 0.2 M) to remove all calcium from the sensors. After analysis, interesting colonies were picked and grown for further investigation from the agar plates with spotted colonies.

## Cell culture and transfection

HeLa cells (CCL-2) acquired from the American Tissue Culture Collection were maintained in full medium, Dulbecco's modified Eagle medium + GlutaMAX (61965, Gibco) supplemented with 10% fetal bovine serum (FBS, 10270, Gibco), under 7% humidified $CO_2$ atmosphere at 37°C. Cells were washed with Hank's buffered salt solution (HBSS, 14175, Gibco) and trypsinized (25300, Gibco) for passaging. No antibiotics were used.

HeLa cells were grown in 24-well plates with a glass bottom for imaging (Thermo Fisher Scientific). Transfection mixture was prepared in Opti-MEM (31985047, Thermo Fisher Scientific) with 2 µg Polyethylenimine in water (PEI, pH 7.3, 23966, Polysciences), 50 ng plasmid DNA and 150 ng mock-DNA (empty plasmid), and incubated for 20 min before addition to the cells. Cells were imaged 1 or 2 days post-transfection. Before imaging, the medium was replaced with microscopy medium (137 mM NaCl, 5.4 mM KCl, 1.8 mM $CaCl_2$, 0.8 mM $MgSO_4$, 20 mM D-glucose, 20 mM HEPES pH 7.4) and cells were incubated for 20 min.

The HeLa cells were from a commercial provider that confirmed the identity of the cell line.

Cord blood Blood Outgrowth Endothelial cells (cbBOEC) were cultivated from healthy donor umbilical cord blood as described before (*Martin-Ramirez et al., 2012*). Cells were cultured in Endothelial Cell Growth Medium-2 BulletKit (CC-3162, Lonza) supplemented with 100 U/ml penicillin (Thermo Fisher Scientific) and 100 µg/ml streptomycin (Thermo Fisher Scientific) and 20% FBS, under 5% humidified $CO_2$ atmosphere at 37°C. cbBOECs were cultured from passage number 4–16. Culture dishes and microscopy plates were coated with 0.1% gelatin (Merck) in phosphate-buffered saline 30 min prior to cell seeding.

Cells were transfected by microporation with 2 µg plasmid DNA containing G-Ca-FLITS tagged to various organelles per 3–5 x $10^5$ cells. For microporation, the Neon Transfection System (MPK5000, Invitrogen) and corresponding Neon transfection kit were used according to manufacturers' protocol. We used the R buffer from the kit, the 100 µl tips, and a 30-ms pulse of 1300 V. After microporation,

BOECs were directly seeded in 24-well plates with a glass bottom (Thermo Fisher Scientific). After 2 hr, the medium was replaced. Cells were imaged 24 hr after transfection in full medium.

The morphology, growth characteristics, and response to histamine (known to increase intracellular calcium) were consistent with those of endothelial cells.

All cell lines were routinely tested for *Mycoplasma* contamination.

## Frequency-domain lifetime imaging

For FD-FLIM lifetime measurements, we used a Lambert Instruments FLIM Attachment (LIFA) setup, composed of an Eclipse Ti microscope (Nikon) with a Lambert Instruments Multi-LED for excitation, a LI²CAM camera, a LIFA signal generator (all Lambert Instruments) to synchronize the light source and the camera, and was controlled by the LI-FLIM software (version 1.2.13). For GFP measurements, a 446 nm light emitting diode (LED) was used, combined with a 448/20 nm or 472/30 nm excitation filter, a 488 nm dichroic mirror, and a 520/35 nm band-pass filter. For CFP measurements, a 446 nm LED was used, combined with a 448/20 nm excitation filter, a 442 nm dichroic mirror, and a 482/25 nm band-pass filter (all filters from Semrock). The LED and image intensifier were high-frequency modulated at 40 MHz. Alexa488 or EB was used as a reference to calibrate the instrumentation, with a known mono-exponential lifetime of 4.05 ns (*Rusinova et al., 2002*; *Zheng et al., 2018*) or 0.086 ns (*Bastiaens et al., 1992*; *Boens et al., 2007*; *van Munster and Gadella, 2004*), respectively.

Fluorescence lifetime of sensors in bacterial lysates was recorded at RT or 37°C using a ×20 (Plan Apo, NA 0.95 air) objective or a ×40 (Plan Apo, NA 0.95 air) objective, collecting 12 phase images and averaging 3×. When measuring lysates in a 96 well-plate, the LIFA software was controlled by a Matlab script (*Bindels et al., 2014*) that automatically moves to the position of each of the wells, adjusts the exposure time based on the intensity to a maximum of 500 ms, collects the lifetime stack, and saves the data. Fluorescence lifetime in HeLa cells or BOECs was recorded at 37°C. If needed, 1 µM histamine or a mix of ionomycin (5 µg/ml, I-6800, LClaboratories) and calcium (5 mM) was added. Cells were imaged using a ×40 (Plan Apo, NA 0.95 air) objective and collecting 12 phase images.

Recorded sample stacks and a reference stack were converted into lifetime images by an ImageJ macro (*Bindels et al., 2020*; *Gadella et al., 1994*). When the intensity of fluorescence in the cells was very weak, a background correction was performed on the lifetime stacks, using a manually indicated background region using Matlab (R2015a). For bacterial lysates, the average phase and modulation lifetime ($\tau_\varphi$ and $\tau_M$) of the full view were extracted by the imageJ macro described above. For cells, the lifetime of individual cells was collected. This was done in a semi-automatic manner, using first an ImageJ macro that masks the image based on intensity, followed by a macro that guides the measurements of the cells and the saving of the ROIs. For signal-to-noise calculations, a rectangular ROI was drawn manually inside cells with an area of 400 pixels (20 × 20, 10 × 40, or 5 × 75 pixels), from which the mean and standard deviation were extracted.

## Protein isolation

Plasmids were prepared for isolation of the proteins mTq2_T203Y, G-Ca-FLITS, and intermediate variant Tq-Ca-FLITS_T203Y. To this end, mScarlet-I, the anti-FRET linker, and the P2A site were removed from the pFR plasmids containing the proteins of interest by digestion with NheI followed by ligation. The new plasmids were named pFPO-mTq2_T203Y, pFPO-G-Ca-FLITS, and pFPO-Tq-Ca-FLITS_T203Y (FPO being an abbreviation for Franka Protein Only) and correct removal was verified by sequencing.

*E. cloni* bacteria containing a pFPO plasmid were grown overnight in 50 ml SKR medium at 37°C while shaking. The next morning, the cultures were put at 20°C for 6 hr, while shaking. Bacteria were harvested by centrifugation for 30 min at 3220 × *g* (5810 R, Eppendorf) at 4°C and washed in 20 ml ice-cold ST buffer (20 mM Tris, 200 mM NaCl, pH 8.0), followed by a second centrifugation. The bacteria were resuspended in 5 ml ST buffer and stored at –20°C.

Later, the bacteria were thawed on ice and 5 mg Lysozyme, 1 µl Benzonase (25 U/µl), and 50 µl Halt Protease Inhibitor were added (10320015, Thermo Fisher Scientific). From now on, all steps were performed on ice or at 4°C. The mixture was incubated for at least 30 min followed by addition of 50 µl of 20% Nonidet-P40. The solution was centrifuged for 30 min at 40,062 × *g* (Sorval Lynx 6000, Thermo Fisher Scientific) and loaded on 1 ml (bed volume) freshly prepared Ni-NTA resin, according to manufacturers' instructions (88221, Thermo Fisher Scientific). After 1-hr incubation while gently

rotating, the beads were washed seven times with 14 ml ST buffer, each time collecting the beads by centrifugation for 2 min at 700 × g. The proteins were eluted once with 0.5 ml ST buffer with 0.6 M imidazole, and a second and third time with 0.5 ml ST buffer with 0.2 M imidazole.

Salt was removed from the isolated proteins by PD10 desalting columns (17-0851-01, Cytiva), according to manufacturers' instructions. Small salts and molecules (imidazole) flow faster through the column than the large proteins. The columns were equilibrated with 10 mM Tris-HCl pH 8.0 before loading, and proteins were eluted with the same buffer. The visually brightest fractions were collected. For storage, aliquots were snap-frozen in liquid nitrogen and stored at –80°C.

## Quantum yield

The QY was determined by measuring the absorbance and emission spectra of a range of dilutions of purified G-Ca-FLITS or Tq-Ca-FLITS_T203Y in 1 ml 0 or 39 µM free $Ca^{2+}$ buffer of Calcium Calibration Buffer Kit #1 (C3008MP, Thermo Fisher Scientific). In addition, mTq2_T203Y was measured diluted in a 10-mM Tris-HCl buffer (pH 8.0). First, a quartz cuvette was filled with 1 ml of the required buffer and the absorbance spectrum (250–550 nm, 1 nm step size) was measured using a spectrophotometer (Libra S70, Biochrom). The cuvette was transferred to a JASCO FP-8500 spectrofluorometer and the emission spectrum was recorded at 460–700 nm with a 2.5 nm bandwidth and 1-nm intervals, while exciting at 450 nm with 2.5 nm bandwidth, using a scan speed of 500 nm/min, 1 s response time, and a medium sensitivity. A small amount of purified protein was added, and the absorbance and emission spectra were recorded again. This was repeated for a total of 4 dilutions per protein, and the absorbance at 450 nm was always below 0.05 for the highest concentration. An additional absorbance spectrum was recorded with a peak absorbance '$A_{peak}$' of 0.5. The same procedure was followed for rhodamine as a reference in 1 mM NaOH.

Absorbance spectra were corrected by subtraction of the offset of the spectrum between 541 and 550 nm, followed by subtraction of the spectrum of only buffer. The resolution of the spectrophotometer is too low for the low protein levels, so the resulting spectra are not smooth. Therefore, the additionally taken absorbance spectrum with $A_{peak}$ = 0.5 were fitted to the low-resolution spectra. From this fit, the absorbance at 450 nm ($A_{450}$) was determined for each dilution of protein. Emission spectra were corrected for spectral sensitivity of the detector and the spectrum of the only buffer sample was subtracted. The spectral area '$I_{em}$' under corrected emission spectra was calculated by integration between 460–700 nm.

The '$A_{450}$' was plotted versus '$I_{em}$' and the slope '$s$' was determined while forcing the regression line through the origin: $A_{450} = s \times I_{em}$. The regression was performed in R Studio (version 1.0.136), using the linear model from the stats package with default parameters. The quantum yield ($QY_s$) of the protein samples was determined according to:

$$QY_s = QY_r \times \frac{s_s}{s_r} \tag{1}$$

where subscripts '$s$' and '$r$' indicate the protein sample and the reference rhodamine, $QY_r$ = 0.85 (*Zhang et al., 2014*), respectively. The standard deviation in '$QY_s$' was determined from the standard deviation of '$s_s$'.

## Extinction coefficient

Purified G-Ca-FLITS, Tq-Ca-FLITS_T203Y, and mTq2_T203Y were ~10× diluted in calcium buffers containing 0 or 39 µM free calcium of the Calcium Calibration Buffer Kit #1 (C3008MP, Thermo Fisher Scientific) to a final volume of 200 µl. The absorbance spectra were measured before and after addition of 0.5 M NaOH and 2 M urea, at 260–650 nm with 1 nm step size and 1 nm bandwidth. The corresponding buffer was used as a reference. The concentration of unfolded protein was determined using the Beer–Lambert law and assuming an extinction coefficient ($\varepsilon$) at 462 nm of 46 $mM^{-1}$ $cm^{-1}$ for the free cyan chromophore (*Lelimousin et al., 2009*). Next, $\varepsilon_{max}$ was determined at maximum absorbance for G-Ca-FLITS, Tq-Ca-FLITS_T203Y, and mTq2_T203Y.

## In vitro calibration

Purified G-Ca-FLITS were diluted 1000× in calcium buffers ranging from 0 to 39 µM, using the Calcium Calibration Buffer Kit #1 according to manufacturers' instructions. Dilutions were made in triplicate

in a 96-well plate with black walls and flat glass bottom (89626, Ibidi). Fluorescence lifetime was recorded of each well as described under 'lifetime imaging', first at RT and later at 37°C, using the same plate. The average phase and modulation lifetime ($\tau_\varphi$ and $\tau_M$) of the full view were determined per well. The average $\tau_\varphi$ and $\tau_M$ were determined for 0 and 39 µM free $Ca^{2+}$, these are 'min' and 'max'. The phase ($\Phi$) and modulation ($M$) were calculated from the recorded lifetimes and displayed in a polar plot as $G$ and $S$ coordinates:

$$M = \sqrt{\frac{1}{1 + \left(\omega\,\tau_M\right)^2}} \text{ and}$$

$G = M\cos\left(\Phi\right)$ and $S = M\sin\left(\Phi\right)$ (2)

The angular frequency of modulation, $2\pi f$, is given by '$\omega$', and we used $f = 40$ MHz. Measurements were projected on the straight line between the two extremes ('min' and 'max') and converted to line fraction '$a$' (**Equation 3**),

$$dS = S - S_{\min} \text{ and}$$

$$a = \frac{dG \times dG_{\max} + dS \times dS_{\max}}{dG_{\max}^2 + dS_{\max}^2} \tag{3}$$

The line fraction was corrected for the intensity contribution of the two states to find the true fraction '$F$', with $F = 1$ representing all sensors in the calcium-bound state (**Equation 4**). The intensity ratio ($R$) between states in vitro was determined to be $R = 0.836$, based on the theoretical ratio calculated from the spectra of the sensor and the instrument settings, see Appendix 2,

$$F = \frac{a}{R \times \left(1 - a\right) + a} \tag{4}$$

The fraction '$F$' was fitted with the Hill equation (**Equation 5**) to find the in vitro $K_d$, using the Nonlinear Least Squares method of the R Stats Package (version 3.3.3) in R Studio (version 1.0.136), using the port algorithm.

$$F = F_{\min} + \frac{\left(F_{\max} - F_{\min}\right)}{\left(\dfrac{K_d}{L}\right)^n + 1} \tag{5}$$

The microscopic dissociation constant is given by '$K_d$', the known free $Ca^{2+}$ concentration by '$L$' and the Hill coefficient by '$n$'.

The 95% confidence interval (CI) of the fraction of the two extremes was determined from the data. The mean of the low calcium state plus the CI was determined to be the lowest measurable fraction, the mean of the high calcium state minus the CI was determined to be the highest measurable fraction. From this, the lowest and highest measurable concentration was determined using the Hill equation (**Equation 5**).

## pH sensitivity

A series of buffers ranging from pH 2.8–10.1 were prepared, using 50 mM citrate buffer (pH 2.8–5.8), MOPS buffer (pH 6.3–7.9) and glycine/NaOH buffer (pH 8.2–10.1). Buffers additionally contain 0.1 M KCl and 0.1 mM $CaCl_2$ or 5 mM EGTA. The pH of each buffer was determined including all components shortly before use. Purified G-Ca-FLITS or mTq2_T203Y was diluted 520× in the prepared buffers, using a predilution of 40 × and a total volume of 200 µl. The dilutions were made in triplicate in a 96-well plate with black walls and flat glass bottom (89626, ibidi). The lifetime of each well in the same 96-well plate was recorded at RT, see 'Lifetime imaging'.

## Ratiometric imaging

HeLa cells co-expressing a green calcium sensor (or green FP) and mScarlet-I form a single plasmid (pFR) were imaged 24 hr after transfection at 37°C, using an Eclipse Ti microscope (Nikon) equipped with a Spectra X Light Engine (Lumencor) for excitation and an Orca flash 4.0 camera (Hamamatsu). Cells were imaged using a Plan Apo ×10 NA 0.45 air objective. For imaging of GFP, a 470 nm LED,

a 470/24 nm excitation filter, and a quad band cube (MXU 71640, Nikon) were used. For the RFP, a 575 nm LED, a 575/25 nm excitation filter, and the same quad band cube were used. Only the middle of the field of view was used to minimize the unequal illumination. A 4 x 4 tile scan was performed and channels were imaged sequentially.

Fluorescence was recorded without stimulation and directly after stimulation with a mix of ionomycin (5 µg/ml, I-6800, LClaboratories) and calcium (5 mM). Background was subtracted from the images and the average green and red fluorescence intensity of individual cells was measured using ImageJ (version 1.52k). The ratio GFP/RFP was calculated for each cell. As a control, we used a sample with only mScarlet-I (ratio GFP/RFP = 0) and a sample with EGFP and mScarlet-I (ratio GFP/RFP = 1). The ratio of all cells was normalized to the controls. The experiment was performed twice per construct.

## Confocal imaging
Confocal images of HeLa cells and BOECs expressing G-Ca-FLITS were taken with a Leica Sp8 system (Leica) with a ×63 objective (HL PL Apo, C2S NA 1.40 oil) at 37°C and 7% (HeLa cells) or 5% $CO_2$ (BOECs). HeLa cells were imaged in MM and BOECs in growth medium, as stated before. A 488 nm Argon laser was used for imaging, combined with an Acousto-Optical Beam Splitter (AOBS), and fluorescence was collected with a HyD at 500–780 nm. The pinhole was set to 1 AU.

## Confocal TCSPC imaging – Picoharp TCSPC module
Confocal fluorescence lifetime images of G-Ca-FLITS expressed in mitochondria of HeLa cells were acquired at an FV1000 (Olympus) confocal microscope equipped with a Picoharp TCSPC module (PicoQuant). A field of view of 256 x 256 pixels (207–331 nm per pixel) was illuminated with a pulsed 485 nm Picoquant diode laser (20 MHz, 0.4 kW cm$^{-2}$) using an Olympus UPLS Apo 60 x water NA1.2 objective lens. The fluorescence signal was detected in confocal mode with the pinhole diameter set at 130 µm. The fluorescence passed a 405/480/560/635 dichroic mirror, was filtered by a 505–540 nm emission filter, and detected by avalanche photodiodes (MPD).

In order to obtain a reliable fluorescence lifetime, the measurement times of the sample channel were set such that at least $10^5$ photons were collected. An image of the median arrival times per pixel was inspected for heterogeneity. Based on the image, one or more regions were manually selected for fitting of a decay curve. The full fluorescence decay curves (5–6 min integration time) were fitted in the Symphotime64 software (PicoQuant) using a bi-exponential decay model including an instrumental response function generated from the same dataset. On the basis of visual inspection of the fit, the fit residuals and the minimal Chi square, the fitted results were accepted or discarded. The fitting of the decay curves was repeated for all areas, but now an integration time of 1 min was used. The fitting was repeated for every minute to obtain information about temporal variation.

## Confocal TCSPC imaging – Leica Stellaris
Confocal fluorescence lifetime images of G-Ca-FLITS expressed in mitochondria of HeLa cells were acquired with a Leica Stellaris8 equipped with a pulsed white light laser set to 40 MHz and a 40x NA 0.95 air PL APO objective. The excitation was at 474 nm and the emission was detected between 490 and 600 nm by the HyDX2 detector. The pinhole was set to 4.4 airy units, the imaging speed to 400 Hz per line (pixel dwell time 3.16 µs) with eight line repetitions and a frame size of 512 x 512 (pixel size 0.259 µm). For live-cell imaging, the images were acquired every minute.

The data was saved as a .ptu file and a custom Python script was used to convert these data into G and S images. The G and S data, together with the $K_D$ determined by the calibration, were used to convert the data into calcium concentration. Code is available (*Goedhart and van de Linden, 2025*): https://github.com/JoachimGoedhart/G-Ca-FLITS (copy archived at *Goedhart, 2025*).

## Imaging and analysis of 2P-FLIM of *Drosophila*
We imaged the brain of *D. melanogaster* females attached to a custom physiology platform (*Maimon et al., 2010*; *Mussells Pires et al., 2024*). This platform keeps the fly's body dry and in air, with the brain exposed to an experimentally determined saline. The flies were freely dangling, and we did not track their behavior. We removed the superior cuticle of the head (the frons and anterior ocellar cuticle) to access the brain dorsally for optical imaging.

Extracellular saline was prepared as reported (**Mussells Pires et al., 2024**), consisting of (in mM): 103 NaCl, 3 KCl, 5 TES, 10 trehalose dihydrate, 10 glucose, 2 sucrose, 26 $NaHCO_3$, 1 $NaH_2PO_4$, 1.5 $CaCl_2$, and 4 $MgCl_2$ titrated with Milli-Q water to an osmolarity of 280–285 mOsm. The high [K+] solution exchanged the concentrations of NaCl with KCl, keeping the osmolarity and $[Cl^-]$ unchanged.

The saline was kept at ~21°C for the duration of all experiments using a Warner Instruments temperature controller (CL-100) driving a Peltier device (SC-20). The Peltier device was placed in-line with tubing (Tygon 2375) that perfused the preparation with saline at a rate of ~3 ml per minute. A thermistor in the bath was used to regulate the Peltier device in closed loop.

The high [K$^+$] solution was applied by allowing flow through the perfusion line to exchange with the extracellular solution, which was gradually aspirated out of the chamber using a glass pipette attached to a vacuum. Each recording was given ~10 min to respond to the new solution, and we estimate that the saline composition of the preparation was fully replaced within ~1 min.

The 2P-FLIM setup uses an Olympus 20x 1.0 NA water immersion objective (XLUMPLFLN) on a customized Sutter MIMMs to be detailed in a later manuscript by SCT and GM. A Coherent Discovery NX laser's tunable beam set to 920 nm (with –12,500 fs2 of dispersion compensation) was used to excite all three fluorophores.

Frames were collected at a resolution of 256x256 pixels and the time series data shown in **Figure 9** are plotted at ~1 Hz. Flies expressing jGCaMP7f or Tq-Ca-FLITS were imaged with ~20 mW of laser power measured after the objective, while flies expressing G-Ca-FLITS were imaged with ~30 mW laser power.

The fluorescence lifetime was computed by fitting the first 1200 frames to a biexponential distribution convolved with a Gaussian plus randomly distributed uniform noise by minimizing the chi-squared statistic (**Thornquist et al., 2020**). The offset of this fit was used to adjust the mean photon arrival time within each frame (empirical lifetime), and the contribution of the estimated noise was subtracted out. Explicitly, the arrival time histogram was fit to the equation:

$$p(t) = \epsilon + f_1 F_1(t; \tau_1, \sigma, \mu) + f_2 F_2(t; \tau_2, \sigma, \mu) \tag{6}$$

where

$$F(t; \tau, \sigma, \mu) = \sqrt{\frac{\pi\sigma^2}{2\tau^2}} \exp\left(\frac{\sigma^2}{2\tau^2}\right) \text{erfc}\left(\frac{1}{\sqrt{2}}\left(\frac{\sigma}{\tau} - \frac{t-\mu}{\sigma}\right)\right) \exp\left(-\frac{t-\mu}{\tau}\right) \tag{7}$$

is the single exponential response with a time constant $\tau$ convolved with a Gaussian centered at μ with standard deviation $\sigma$ and $\epsilon + f_1 + f_2 = 1$. To compute the empirical lifetime $L$ of a collection of photons, we computed the mean arrival time of the photons $\langle p \rangle = \sum_{t=0}^{t_{max}} t * p(t)$, then removed the offset and the estimated noise due to the uniform signal (which had mean arrival time $\frac{t_{max}}{2}$ because it is uniformly distributed):

$$L = \langle p \rangle - \left(\mu + \epsilon \frac{t_{max}}{2}\right). \tag{8}$$

Mean intensity-weighted lifetime refers to the following procedure: when taking the mean of a set of $n$ frames, the lifetime was computed as

$$\langle L \rangle = \sum_{k=1}^{n} \frac{I_k L_k}{n \langle I \rangle} \tag{9}$$

where $I_k$ refers to the intensity of the $k$ th frame, $L_k$ refers to the mean lifetime of the $k$ th frame, and $\langle \circ \rangle$ refers to the average across frames.

## Acknowledgements

FHL was supported by a NWO Chemical Sciences ECHO grant (711.017.003). GM is a Howard Hughes Medical Institute Investigator. The funder(s) had no role in study design, data collection, and analysis, decision to publish, or preparation of the manuscript. We thank Ronald Breedijk (University of

Amsterdam) for keeping our microscopes in excellent condition and particularly for help with 2P imaging of purified protein.

## Additional information

### Funding

| Funder | Grant reference number | Author |
|---|---|---|
| NWO Chemical Sciences | ECHO grant (711.017.003) | Franka H van der Linden |

The funders had no role in study design, data collection, and interpretation, or the decision to submit the work for publication.

### Author contributions

Franka H van der Linden, Conceptualization, Supervision, Investigation, Visualization, Methodology, Writing – original draft, Project administration, Writing – review and editing; Stephen C Thornquist, Resources, Formal analysis, Investigation, Visualization, Methodology, Project administration, Writing – review and editing; Rick M ter Beek, Jelle Y Huijts, Investigation, Methodology, Writing – review and editing; Mark A Hink, Formal analysis, Investigation, Writing – review and editing; Theodorus W J Gadella, Conceptualization, Software, Supervision, Investigation, Methodology, Writing – review and editing; Gaby Maimon, Supervision, Writing – review and editing; Joachim Goedhart, Conceptualization, Funding acquisition, Visualization, Methodology, Writing – original draft, Project administration, Writing – review and editing

### Author ORCIDs

Franka H van der Linden  https://orcid.org/0000-0001-5324-2714
Stephen C Thornquist  https://orcid.org/0000-0003-2693-4971
Theodorus W J Gadella  https://orcid.org/0000-0002-7639-219X
Gaby Maimon  https://orcid.org/0000-0003-1219-5856
Joachim Goedhart  https://orcid.org/0000-0002-0630-3825

Reviewer #1 (Public review): https://doi.org/10.7554/eLife.105086.3.sa1
Reviewer #2 (Public review): https://doi.org/10.7554/eLife.105086.3.sa2
Reviewer #3 (Public review): https://doi.org/10.7554/eLife.105086.3.sa3
Author response https://doi.org/10.7554/eLife.105086.3.sa4

## Additional files

### Supplementary files

Supplementary file 1. Lifetime and spectral screen of candidate sensors to create G-Ca-FLITS.

Supplementary file 2. Primers.

MDAR checklist

### Data availability

Several plasmids are deposited for distribution through Addgene (https://www.addgene.org/). The plasmids and corresponding Addgene numbers are: N1-G-Ca-FLITS: RRID:Addgene_191465, 3xnls-G-Ca-FLITS: RRID:Addgene_191463, 4xmts-G-Ca-FLITS: RRID:Addgene_191462, ER-G-Ca-FLITS-KDEL: RRID:Addgene_191464, pFPO-His-mTurquoise2_T203Y: RRID:Addgene_191456, pFPO-His-G-Ca-FLITS: RRID:Addgene_191455, pFR-mScarletI-antiFRET-P2A-G-Ca-FLITS: RRID:Addgene_191457, pFR-mScarletI-antiFRET-P2A-EGFP: RRID:Addgene_191458, pFR-mScarletI-antiFRET-P2A-mTurquoise2_T203Y: RRID:Addgene_191459, pFR-mScarletI-antiFRET-P2A-GCaMP3:RRID:Addgene_191460, pFR-mScarletI-antiFRET-P2A-jGCaMP7c: RRID:Addgene_191461, pDress-mScarletI-antiFRET-P2A2-linker_noSacI: RRID:Addgene_191473. Transgenic Drosophila lines expressing Tq-Ca-FLITS under UAS or LexA control and G-Ca-FLITS under UAS control are available

upon request. Data is available at zenodo, DOI: 10.5281/zenodo.17237551. Code is available at Github: https://github.com/JoachimGoedhart/G-Ca-FLITS (copy archived at *Goedhart, 2025*).

The following dataset was generated:

| Author(s) | Year | Dataset title | Dataset URL | Database and Identifier |
|---|---|---|---|---|
| van der Linden F, Goedhart J | 2025 | Data related to the publication "A green lifetime biosensor for calcium that remains bright over its full dynamic range" | https://doi.org/10.5281/zenodo.17237551 | Zenodo, 10.5281/zenodo.17237551 |

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

# Appendix 1

## A high-throughput method to screen sensors in bacterial lysate

Before creating sensor variants, we developed a high-throughput method to screen different calcium sensors in bacterial lysate. The aim was to grow and isolate 96 bacterial cultures in parallel in a deep well plate. A previously developed lysis buffer (*Danilevich et al., 2008*) served as a starting point for optimization of the protocol. Bacteria expressing the sensor Tq-Ca-FLITS were used to compare the lysis efficiency. The following variables were tested in a 50 mM Tris-HCl buffer: (1) presence or absence of a freeze/thawing cycle, (2) presence or absence of lysozyme, (3) different concentrations of urea (0, 0.5, and 1 M), and (4) different concentrations of deoxycholic acid (DOC, 1, 2, and 3%). After lysis, the fluorescence intensity of the bacterial lysates was compared (*Appendix 1—figure 1*).

In general, a freezing and thawing cycle resulted in a higher fluorescence intensity for most conditions. However, this doubled the duration of the protocol. A high concentration of DOC and urea combined was too harsh for the fluorescent sensor: no intensity was detected. All other conditions were comparable. We favored a buffer without urea to maintain the native state of the proteins, and without lysozyme to omit the need to freshly add this component before each use. Therefore, we chose a protocol with 2% DOC and without a freezing step. The obtained lysates were of sufficient brightness and purity to use for spectral measurements and lifetime measurements.

We explored the possibility of replacing the 2% DOC with other detergents, but no improvements were found. Triton X-100 (0.1–1%) gave lower intensities compared to DOC and caused a problematic amount of foam during handling. Sodium dodecyl sulfate (SDS, 0.1–0.5%), an anionic surfactant in contrast to non-ionic Triton X-100 and DOC, also gave lower intensities and poses the risk of denaturation of the proteins (*Appendix 1—figure 2*). Therefore, we chose the 50 mM Tris-HCl buffer with 2% DOC for further experiments.

The lysates were screened for a lifetime change and a change in excitation and emission spectra, measured in both the calcium-bound and calcium-free states by addition of CaCl2 or EDTA (*Appendix 1—figure 3*).

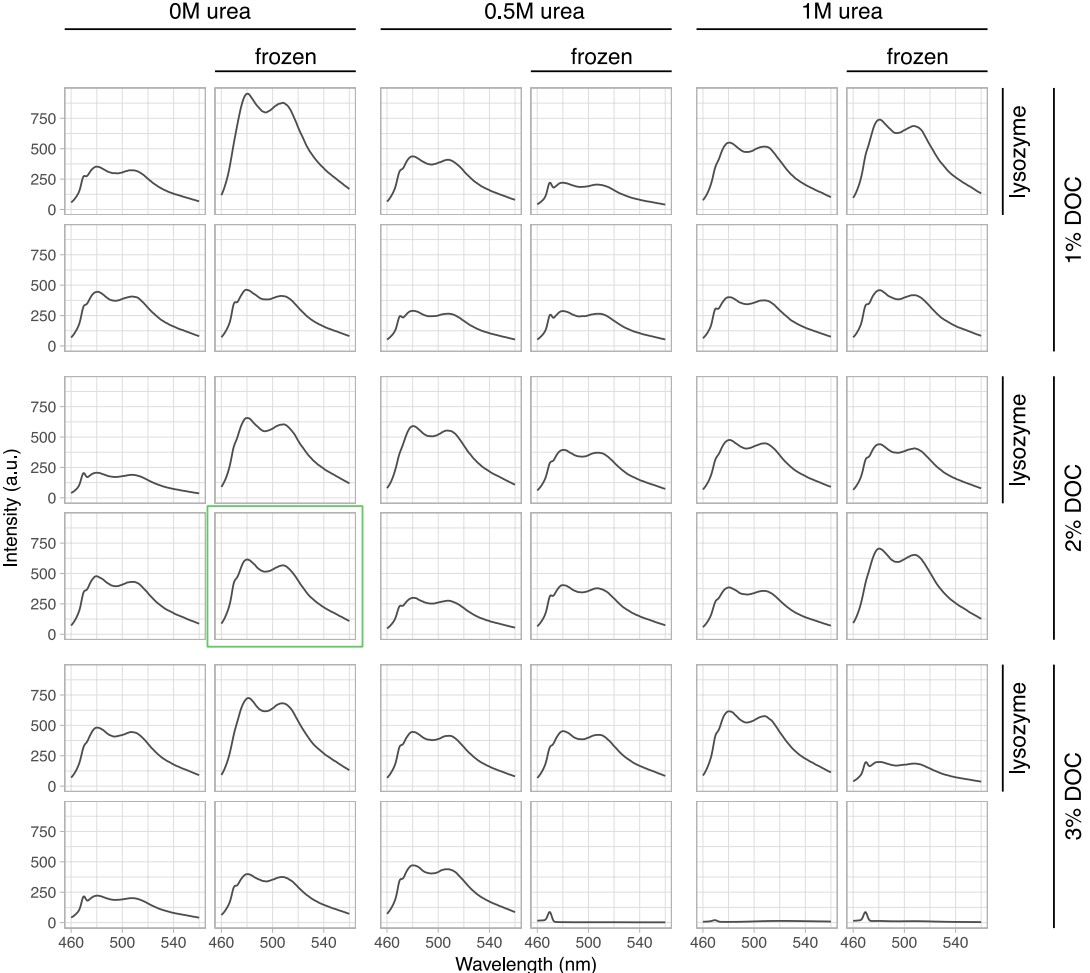

**Appendix 1—figure 1.** Influence of different components of a bacterial lysis protocol. Emission spectra of bacterial lysates expressing Tq-Ca-FLITS are shown. Various concentrations of urea, DOC, and lysozyme were used in the lysis buffer. Each buffer also contained 50 mM Tris-HCl (pH 8.0). The influence of a freeze/thawing step is also tested, indicated by 'frozen'. Each condition was measured once. The condition that was selected for the screening is indicated with a green box.

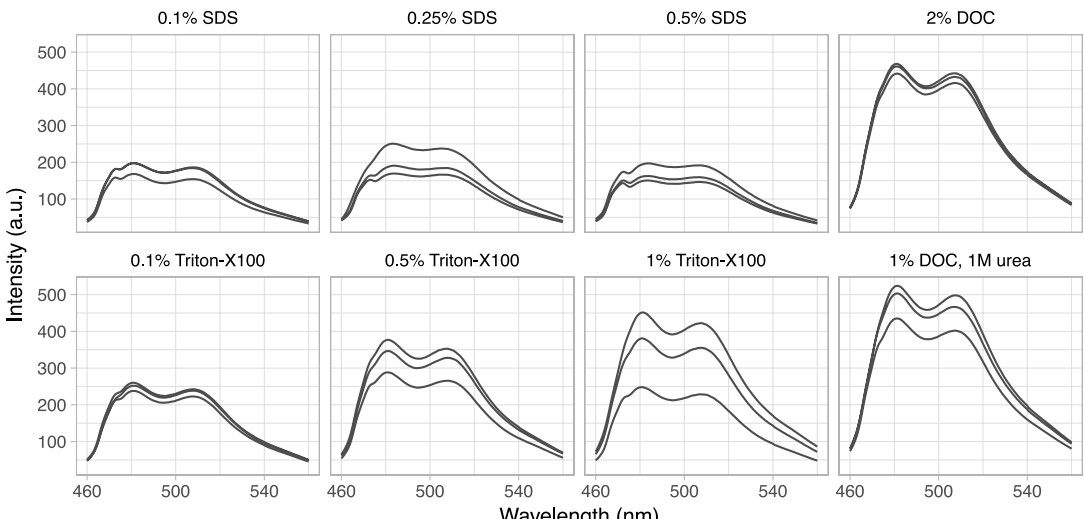

**Appendix 1—figure 2.** Influence of different detergents on a bacterial lysis protocol. Emission spectra of bacterial lysates expressing Tq-Ca-FLITS are shown. Various concentrations of SDS and Triton X-100 were tested and compared to a buffer with 2% DOC and a buffer with 1% DOC and 1 M urea, the original bacterial lysis buffer (*Danilevich et al., 2008*). Each buffer also contained 50 mM Tris-HCl (pH 8.0). Each condition was measured three times.

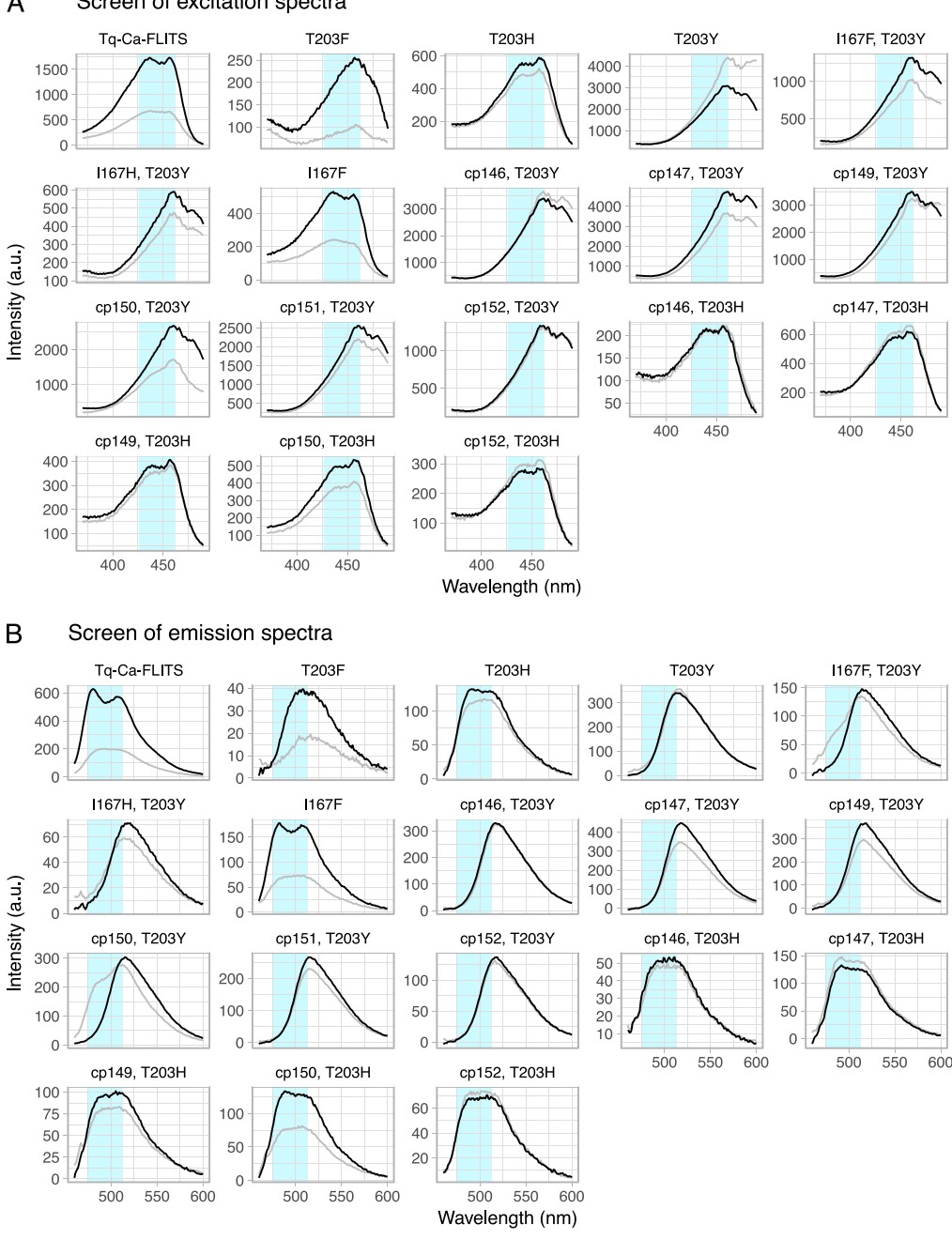

**Appendix 1—figure 3.** Excitation (**A**) and emission spectra (**B**) of screened variants in the search for a red-shifted calcium sensor based on Tq-Ca-FLITS. The spectra were measured in bacterial lysates, with addition of 0.1 mM CaCl$_2$ (black line) or 9.5 mM EDTA (gray line). Sensor variants are indicated by their circular permutation and mutations in the FP in the sensor. For example, 'cp146' has from N- to C-terminus the following domains: calmodulin binding peptide M13, amino acids 146–283 of mTq2, a flexible GGSGG linker, amino acids 1–145 of mTq2, CaM (calmodulin). If no circular permutation position is indicated, the mutation displayed is done on Tq-Ca-FLITS. 'T203Y' indicates a T203Y mutation on the mTq2 in the sensor. The light blue shading is used as a reference for excitation and emission peaks of Turquoise.

## Appendix 2

To determine the fraction of G-Ca-FLITS in the calcium-bound state for the lifetime calibration, the intensity contribution each state needs to be determined for the specific microscope settings used for the calibration. To this end, we applied a theoretical correction of both the extinction coefficient and the quantum yield to obtain a relative brightness per state (***Appendix 2—figure 1***).

The extinction coefficient for the calcium-bound and calcium-free state was calculated over the full absorption spectrum by considering:

1. The normalized brightness of the excitation light (LED) $l(\lambda)$.
2. The normalized transmission of the excitation filter $ExF(\lambda)$.
3. The normalized reflection of the dichroic mirror $DichrR(\lambda)$.
4. The normalized excitation spectra, for the correspondi.ng state of the sensor $F_{ex}(\lambda)$.
5. The maximal extinction coefficient $\varepsilon_{max}$.

The effective extinction coefficient ($\varepsilon_{eff}$) is calculated as:

$$\varepsilon_{eff} = \varepsilon_{max} \frac{\int l(\lambda) \cdot ExF(\lambda) \cdot DichrRF(\lambda) \cdot F_{ex}(\lambda)\, d\lambda}{\int l(\lambda) \cdot ExF(\lambda) \cdot DichrR(\lambda)\, d\lambda}.$$

The relative excitation efficiency ($e_{rel}$) with the deployed excitation source and filters is defined as:

$$e_{rel} = \frac{\varepsilon_{eff}}{\varepsilon_{max}} = \frac{\int l(\lambda) \cdot ExF(\lambda) \cdot DichrR(\lambda) \cdot F_{ex}(\lambda)\, d\lambda}{\int l(\lambda) \cdot ExF(\lambda) \cdot DichrR(\lambda)\, d\lambda}.$$

The corrected emission spectra ($F_{em}(\lambda)$) of both the calcium-free and the calcium-bound state were corrected by considering:

1. The normalized transmission of the dichroic mirror $DichrT(\lambda)$.
2. The normalized transmission of the emission filter $EmF(\lambda)$.
3. The normalized sensitivity of the camera $S(\lambda)$.
4. The quantum yield of the sensor QY.

From these the relative sensitivity ($S_{rel}$) for detection of the sensor can be computed:

$$S_{rel} = \frac{\int EmF(\lambda) \cdot DichrTF(\lambda) \cdot S(\lambda) \cdot F_{em}(\lambda)\, d\lambda}{\int F_{em}(\lambda)\, d\lambda}.$$

Consequently, the relative overall measured brightness ($B_{rel}$) for a specific combination of optical components is defined as:

$$B_{rel} = e_{rel}\varepsilon_{max}S_{rel}\text{QY}.$$

Applying this calculation to the spectra of G-Ca-FLITS using the settings throughout this manuscript (LED centered around 446 nm, excitation filter ff01-448/20, dichroic di02-r488-25, emission filter ff01-520/35), results in an $\varepsilon_{eff}$ of 15,900 and 21,300 M$^{-1}$ cm$^{-1}$ for the calcium-free and -bound states, respectively. The $S_{rel}$ is 0.41 for both states. Combining $\varepsilon_{eff}$ and $S_{rel}$ with the determined QY and $\varepsilon_{max}$, results in a $B_{rel}$ of 2710 and 2270 M$^{-1}$ cm$^{-1}$ for the calcium-free and -bound states, respectively. Therefore, the intensity contribution of the calcium-free state is 1.2-fold higher than the calcium-bound state.

Using the same calculation for the sensor Tq-Ca-FLITS with suitable settings (LED centered around 446 nm, excitation filter ff01-448/20, dichroic di02-r442-25, emission filter ff01-482/25), results in a 3.6-fold higher intensity contribution of the calcium-bound state compared to the calcium-free state. This is almost identical to the ratio of 3.51 measured and reported previously, where the same setup and settings were applied (***van der Linden et al., 2021***, doi:10.1038/s41467-021-27249-w).

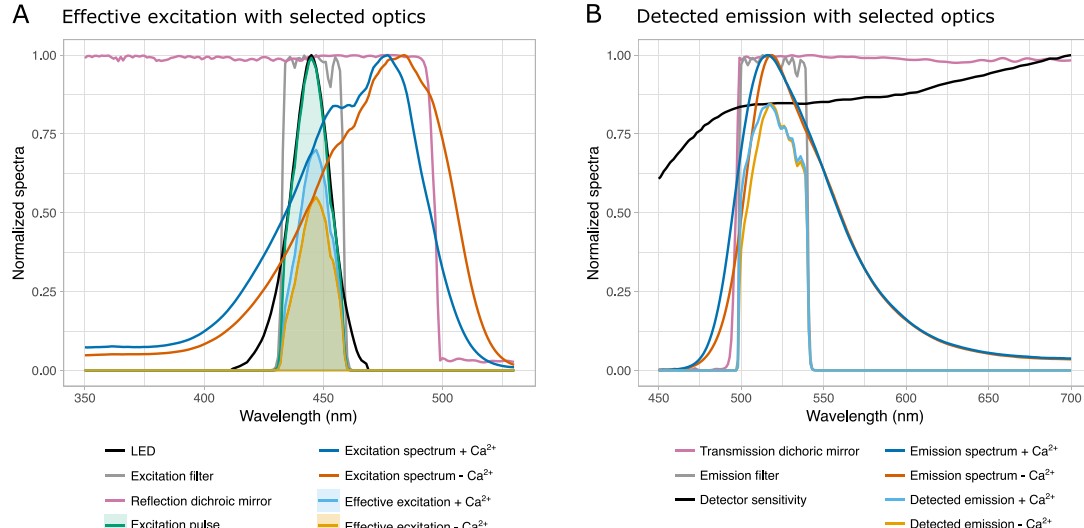

**Appendix 2—figure 1.** Effective excitation and detected emission of G-Ca-FLITS for the indicated filters. Excitation and emission spectra were recorded using isolated protein. Filter profiles and the detector sensitivity are provided on the manufacturers' respective websites. The LED brightness was measured. (**A**) All normalized spectra related to excitation of G-Ca-FLITS are shown. The 'effective excitation pulse' ('LED' × 'excitation filter' × 'reflection dichroic mirror') is shown in green. The 'effective excitation pulse' is multiplied with the 'excitation spectra' to yield the 'effective excitation' of G-Ca-FLITS for each state (cyan and orange). The $e_{rel}$ discussed in Appendix 2 is the integral of 'effective excitation' divided over the integral of 'excitation pulse'. (**B**) All normalized spectra related to emission and detection of G-Ca-FLITS are shown. The 'detected emission' of G-Ca-FLITS was calculated for each state by multiplication of the following spectra: 'transmission dichroic mirror' × 'emission filter' × 'detector sensitivity' × 'emission spectrum'. The $S_{rel}$ discussed in Appendix 2 is the integral of 'detected emission' divided over the integral of 'emission spectra' for each calcium state.

