## [Editor Report · eLife Assessment]

This manuscript reports on an FLIM-based calcium biosensor, G-CaFLITS. It represents an **important** contribution to the field of genetically-encoded fluorescent biosensors, and will serve as a practical tool for the FLIM imaging community. The paper provides **convincing** evidence of G-CaFLITS's photophysical properties and its advantages over previous biosensors such as Tq-Ca-FLITS. Although the benefits of G-Ca-FLITS over Tq-Ca-FLITS are limited by the relatively small wavelength shift, it presents some advantages in terms of compatibility with available instrumentation and brightness consistency.

---

## [Referee Report · Reviewer #1 (Public review)]

Summary:

van der Linden et al. report on the development of a new green-fluorescent sensor for calcium, following a novel rational design strategy based on the modification of the cyan-emissive sensor mTq2-CaFLITS. Through a mutational strategy similar to the one used to convert EGFP into EYFP, coupled with optimization of strategic amino acids located in proximity of the chromophore, they identify a novel sensor, G-CaFLITS. Through a careful characterization of the photophysical properties in vitro and the expression level in cell cultures, the authors demonstrate that G-CaFLITS combines a large lifetime response with a good brightness in both the bound and unbound states. This relative independence of the brightness on calcium binding, compared with existing sensors that often feature at least one very dim form, is an interesting feature of this new type of sensors, which allows for a more robust usage in fluorescence lifetime imaging. Furthermore, the authors evaluate the performance of G-CaFLITS in different subcellular compartments and under two-photon excitation in Drosophila. While the data appears robust and the characterization thorough, the interpretation of the results in some cases appears less solid, and alternative explanations cannot be excluded.

Strengths:

The approach is innovative and extends the excellent photophysical properties of the mTq2-based to more red-shifted variants. While the spectral shift might appear relatively minor, as the authors correctly point out, it has interesting practical implications, such as the possibility to perform FLIM imaging of calcium using widely available laser wavelengths, or to reduce background autofluorescence, which can be a significant problem in FLIM.

The screening was simple and rationally guided, demonstrating that, at least for this class of sensors, a careful choice of screening positions is an excellent strategy to obtain variants with large FLIM responses without the need of high-throughput screening.

The description of the methodologies is very complete and accurate, greatly facilitating the reproduction of the results by others, or the adoption of similar methods. This is particularly true for the description of the experimental conditions for optimal screening of sensor variants in lysed bacterial cultures.

The photophysical characterization is very thorough and complete, and the vast amount of data reported in the supporting information is a valuable reference for other researchers willing to attempt a similar sensor development strategy. Particularly well done is the characterization of the brightness in cells, and the comparison on multiple parameters with existing sensors.

Overall, G-CaFLITS displays excellent properties for a FLIM sensor: very large lifetime change, bright emission in both forms and independence from pH in the physiological range.

Comment on revised version:

The authors have significantly improved the manuscript, and overall I fully agree in maintaining the assessment as it is now.

---

## [Referee Report · Reviewer #2 (Public review)]

Summary:

Van der Linden et al. describe the addition of the T203Y mutation to their previously described fluorescence lifetime calcium sensor Tq-Ca-FLITS to shift the fluorescence to green emission. This mutation was previously described to similarly red-shift the emission of green and cyan FPs. Tq-Ca-FLITS_T203Y behaves as a green calcium sensor with opposite polarity compared with the original (lifetime goes down upon calcium binding instead of up). They then screen a library of variants at two linker positions and identify a variant with slightly improved lifetime contrast (Tq-Ca-FLITS_T203Y_V27A_N271D, named G-Ca-FLITS). The authors then characterize the performance of G-Ca-FLITS relative to Tq-Ca-FLITS in purified protein samples, in cultured cells, and in the brains of fruit flies.

Strengths:

This work is interesting as it extends their prior work generating a calcium indicator scaffold for fluorescent protein-based lifetime sensors with large contrast at a single wavelength, which is already being adopted by the community for production of other FLIM biosensors. This work effectively extends that from cyan to green fluorescence. While the cyan and green sensors are not spectrally distinct enough (~20-30nm shift) to easily multiplex together, it at least shifts the spectra to wavelengths that are more commonly available on commercial microscopes.

The observations of organellar calcium concentrations were interesting and could potentially lead to new biological insight if followed up.

---

## [Referee Report · Reviewer #3 (Public review)]

Summary:

The authors present a variant of a previously described fluorescence lifetime sensor for calcium. Much of the manuscript describes the process of developing appropriate assays for screening sensor variants, and thorough characterization of those variants (inherent fluorescence characteristics, response to calcium and pH, comparisons to other calcium sensors). The final two figures show how the sensor performs in cultured cells and in vivo drosophila brains.

Strengths:

The work is presented clearly and the conclusion (this is a new calcium sensor that could be useful in some circumstances) is supported by the data.

Weaknesses:

There are probably few circumstances where this sensor would facilitate experiments (calcium measurements) that other sensors would prove insufficient.

Comment on revised version:

I think the manuscript has been significantly improved and I concur with the eLife Assessment statement.

[Editors' note: There are no further requests by the reviewers. All of them expressed their approval of the new version of the manuscript.]

---

## [Author Response]

The following is the authors’ response to the original reviews.

**Reviewer #1 (Public review):**
Summary:van der Linden et al. report on the development of a new green-fluorescent sensor for calcium, following a novel rational design strategy based on the modification of the cyan-emissive sensor mTq2-CaFLITS. Through a mutational strategy similar to the one used to convert EGFP into EYFP, coupled with optimization of strategic amino acids located in proximity of the chromophore, they identify a novel sensor, GCaFLITS. Through a careful characterization of the photophysical properties in vitro and the expression level in cell cultures, the authors demonstrate that G-CaFLITS combines a large lifetime response with a good brightness in both the bound and unbound states. This relative independence of the brightness on calcium binding, compared with existing sensors that often feature at least one very dim form, is an interesting feature of this new type of sensors, which allows for a more robust usage in fluorescence lifetime imaging. Furthermore, the authors evaluate the performance of G-CaFLITS in different subcellular compartments and under two-photon excitation in Drosophila. While the data appears robust and the characterization thorough, the interpretation of the results in some cases appears less solid, and alternative explanations cannot be excluded.Strengths:The approach is innovative and extends the excellent photophysical properties of the mTq2-based to more red-shifted variants. While the spectral shift might appear relatively minor, as the authors correctly point out, it has interesting practical implications, such as the possibility to perform FLIM imaging of calcium using widely available laser wavelengths, or to reduce background autofluorescence, which can be a significant problem in FLIM.The screening was simple and rationally guided, demonstrating that, at least for this class of sensors, a careful choice of screening positions is an excellent strategy to obtain variants with large FLIM responses without the need of high-throughput screening.The description of the methodologies is very complete and accurate, greatly facilitating the reproduction of the results by others, or the adoption of similar methods. This is particularly true for the description of the experimental conditions for optimal screening of sensor variants in lysed bacterial cultures.The photophysical characterization is very thorough and complete, and the vast amount of data reported in the supporting information is a valuable reference for other researchers willing to attempt a similar sensor development strategy. Particularly well done is the characterization of the brightness in cells, and the comparison on multiple parameters with existing sensors.Overall, G-CaFLITS displays excellent properties for a FLIM sensor: very large lifetime change, bright emission in both forms and independence from pH in the physiological range.Weaknesses:The paper demonstrates the application of G-CaFLITS in various cellular subcompartments without providing direct evidence that the sensor's response is not affected by the targeting. Showing at least that the lifetime values in the saturated state are similar in all compartments would improve the robustness of the claims.In some cases, the interpretation of the results is not fully convincing, leaving alternative hypotheses as a possibility. This is particularly the case for the claim of the origin of the strongly reduced brightness of G-CaFLITS in Drosophila. The explanation of the intensity changes of G-CaFLITS also shows some inconsistency with the basic photophysical characterization.While the claims generally appear robust, in some cases they are conveyed with a lack of precision. Several sentences in the introduction and discussion could be improved in this regard. Furthermore, the use of the signal-to-noise ratio as a means of comparison between sensors appears to be imprecise, since it is dependent on experimental conditions.

We thank the reviewer for a thorough evaluation and for suggestions to improve our manuscript. We are happy with the recognition of the strengths of this work. The list with weaknesses has several valid points which will be addressed in a point-by-point reply and a revision.

**Reviewer #2 (Public review):**
Summary:Van der Linden et al. describe the addition of the T203Y mutation to their previously described fluorescence lifetime calcium sensor Tq-Ca-FLITS to shift the fluorescence to green emission. This mutation was previously described to similarly red-shift the emission of green and cyan FPs. Tq-Ca-FLITS_T203Y behaves as a green calcium sensor with opposite polarity compared with the original (lifetime goes down upon calcium binding instead of up). They then screen a library of variants attwo linker positions and identify a variant with slightly improved lifetime contrast (TqCa-FLITS_T203Y_V27A_N271D, named G-Ca-FLITS). The authors then characterize the performance of G-Ca-FLITS relative to Tq-Ca-FLITS in purified protein samples, in cultured cells, and in the brains of fruit flies.Strengths:This work is interesting as it extends their prior work generating a calcium indicator scaffold for fluorescent protein-based lifetime sensors with large contrast at a single wavelength, which is already being adopted by the community for production of other FLIM biosensors. This work effectively extends that from cyan to green fluorescence. While the cyan and green sensors are not spectrally distinct enough (~20-30nm shift) to easily multiplex together, it at least shifts the spectra to wavelengths that are more commonly available on commercial microscopes.The observations of organellar calcium concentrations were interesting and could potentially lead to new biological insight if followed up.Weaknesses:(1) The new G-Ca-FLITS sensor doesn't appear to be significantly improved in performance over the original Tq-Ca-FLITS, no specific benefits are demonstrated.(2) Although it was admirable to attempt in vivo demonstration in Drosophila with these sensors, depolarizing the whole brain with high potassium is not a terribly interesting or physiological stimulus and doesn't really highlight any advantages of their sensors; G-Ca-FLITS appears to be quite dim in the flies.

We thank the reviewer for a thorough evaluation and for suggestions to improve our manuscript. Although the spectral shift of the green variant is modest, we have added new data (figure 7) to the manuscript that demonstrates multiplex imaging of G-Ca-FLITS and Tq-Ca-FLITS.

As for the listed weaknesses we respond here:

(1) Although we agree that the performance in terms of dynamic range is not improved, the advantage of the green sensor over the cyan version is that the brightness is high in both states.

(2) We agree that the performance of G-Ca-FLITS is disappointing in Drosophila. We feel that this is important data to report, and it makes it clear that Tq-Ca-FLITS is a better choice for this system. Depolarization of the entire brain was done to measure the maximal lifetime contrast.

**Reviewer #3 (Public review):**
Summary:The authours present a variant of a previously described fluorescence lifetime sensor for calcium. Much of the manuscript describes the process of developing appropriate assays for screening sensor variants, and thorough characterization of those variants (inherent fluorescence characteristics, response to calcium and pH, comparisons to other calcium sensors). The final two figures show how the sensor performs in cultured cells and in vivo drosophila brains.Strengths:The work is presented clearly and the conclusion (this is a new calcium sensor that could be useful in some circumstances) is supported by the data.Weaknesses:There are probably few circumstances where this sensor would facilitate experiments (calcium measurements) that other sensors would prove insufficient.

We thank the reviewer for the evaluation of our manuscript. As for the indicated weakness, we agree that the main application of genetically encoded calcium biosensors is to measure qualitative changes in calcium. However, it can be argued that due to a lack of tools the absolute quantification has been very challenging. Now, thanks to large contrast lifetime biosensors the quantitative measurements are simplified, there are new opportunities, and the probe reported here is an improvement over existing probes as it remains bright in both states, further improving quantitative calcium measurements.

**Reviewer #1 (Recommendations for the authors):**
While the science in the paper appears solid, the methods well grounded and excellently documented, the manuscript would benefit from a revision to improve the clarity of the exposition. In particular:Part of the introduction appears like a patchwork of information with poor logical consequentiality. The authors rapidly pass from the impact of brightness on FLIM accuracy, to mitochondrial calcium in pathology, to the importance of the sensor's affinity, to a sentence on sensor's kinetics, to fluorescent dyes and bioluminescence, to conclude that sensors should be stable at mitochondrial pH. I highly recommend rewriting this part.

We thank the referee for the comment and we have adjusted to introduction to better connect the parts and increase the logic. The updated introduction addresses all the feedback by the reviewers on different aspects of the introductory text, and we have removed the section on dyes and bioluminescence. We feel that the introduction is better structured now.

The reference to particular amino acid positions would greatly benefit from including images of the protein structure in which the positions are highlighted, similar to what the same authors do in their fluorescent protein development papers. While in the case of sensors a crystal structure might be lacking, highlighting the positions with respect to an AlphaFold-generated structure or the structure of mTq2 might still be helpful.

We appreciate this remark and we have added a sequence alignment of the FLITS probes to supplemental Figure S4. This shows the residues with number, and we have also highlighted the different domains, linkers and mutations. We think that this linear representation works better than a 3D structure (one issue is that alphafold fails to display the chromophore and it has usually poor confidence for linker residues).

The use of SNR, as defined by the authors (mean of the lifetime divided by standard deviation) appears a poorly suited parameter to compare sensors, as it depends on the total number of collected photons and on the strength of the algorithms used to retrieve the lifetime value. In an extreme example, if one would collect uniform images with millions of photons per pixel, most likely SNR would be extremely good for all sensors in all states, irrespective of the fact that some states are dimmer (within reasonable limits). On the other hand, if the same comparison would be performed at a level of thousands or hundreds of photons per pixel, the effect of different brightness on the SNR would be much more dramatic. While in general I fully agree with the core concept of the paper, i.e. that avoiding low-brightness forms leads more easily to experiments with higher SNR, I would suggest to stick to comparing the sensors in terms of brightness and refer to SNR (if needed) only when describing the consequences on measurements.

The reviewer is right that in absolute terms the SNR is not meaningful. In addition to acquisition time, it depends on expression levels. Yet, it is possible to compare the change in SNR between the apo- and saturated states, and that is what is shown in figure 5. We have added text to better explain that the change in SNR is relevant here:

“The absolute SNR is not relevant here, as it will depend on the expression level and acquisition time. But since we have measured the two extremes in the same cells, we can evaluate how the SNR changes between these states for each separate probe”

Some statements from the authors or aspects of the paper appear problematic:(1) "Additionally, the fluorescence of most sensors is a non-linear function of calcium concentration, usually with Hill coefficients between 2 and 3. This is ideal when the probe is used as a binary detector for increases in Ca2+ concentrations, but it makes robust quantification of low, or even intermediate, calcium concentrations extremely challenging."To the best of my knowledge, for all sensors the fluorescence response is a nonlinear function of calcium concentrations. If the authors have specific examples in mind in which this is not true, they should cite them specifically. Furthermore, the Hill coefficient defines the range of concentrations in which the sensor operates, while the fact that "low concentrations" might be hard to detect depends only on the dim fluorescence of some sensors in the unbound form.

We agree with the reviewer that this part is not clearly written and confusing, as the sentence “Additionally, the fluorescence of most sensors is a non-linear function of calcium concentration, usually with Hill coefficients between 2 and 3” was not relevant in this section and so we removed it. Now it reads:

“Many GECIs harboring a single fluorescent protein (FP), like GCaMPs, are optimized for a large intensity change, and have a (very) dim state when calcium levels are below the KD of the probe (Akerboom et al., 2013; Dana et al., 2019; Shen et al., 2018; Zhang et al., 2023; Zhao et al., 2011). This is ideal when the probe is used as a binary detector for increases in Ca2+ concentrations, but it makes robust quantification of low, or even intermediate, calcium concentrations extremely challenging”

(2) "The affinity of a sensor is of major importance: a low KD can underestimate high concentrations and vice versa."It is not clear to me why the concentrations would be underestimated, rather than just being less precise. Also, if a calibration curve is plotted in linear scale rather than logarithmic scale, it appears that the precision problem is much more severe near saturation (where low lifetime changes result in large concentration changes) than near zero (where low concentration changes produce large lifetime changes).

We agree that this could be better explained, what we meant to say that concentrations that are ~10x lower or higher than the KD cannot be precisely measured. See also our reply to the next comment.

(3) "Differences can also arise due to the method of calibration, i.e. when the absolute minimum and maximum signal are not reached in the calibration procedure (Fernandez-Sanz et al., 2019)."Unless better explained, this appears obvious and not worth mentioning.

What may be obvious to the reviewer (and to us) may not be obvious to the reader, and that’s why this is included. To make it clearer we rephrased this part as a list of four items:

“Accurate determination of the affinity of a sensor is important and there are several issues that need to be considered during the calibration and the measurements: (i) the concentrations can only be measured with sufficient precision when it is in the range between 10x K_D_ and 1/10x K_D_, (ii) the calibration is only valid when the two extremes are reached during the calibration procedure (Fernandez-Sanz et al., 2019), (iii) the sensor’s kinetics should be sufficiently fast enough to be able to track the calcium changes, and (iv) the biosensor should be compatible with the high mitochondrial pH of 8 (Cano Abad et al., 2004; Llopis et al., 1998).”

(4) In the experiments depicted in Figure 6C the underlying assumption is that the sensor behaves in the same way independently of the compartment to which it is targeted. This is not necessarily the case. It would be valuable to see the plots of Figure 6C and D discussed in terms of lifetime. Is the saturating lifetime value the same in all compartments?

This is a valid point and we have now included a plot with the actual lifetime data for each of the organelles (figure S15).

We have also added text to discuss this point: “We note that the underlying assumption of the quantification of organellar calcium concentrations is that the lifetime contrast is the same. This is broadly true for most of the measurements (Figure S15). Yet, there are also differences. It is currently unclear whether the discrepancies are due to differences in the physicochemical properties of the compartments, or whether there is a technical reason (the efficiency of ionomycin for saturating the biosensor in the different compartments is unknown, as far as we know). This is something that is worth revisiting. A related issue that deserves attention is the level of agreement between in vitro and in vivo calibrations.”

(5) A similar problem arises for the observation of different calcium levels in peripheral mitochondria. In figure S11b, the values of the two lifetime components of a biexponential fit are displayed. Both the long and short components seem to be different. This is an interesting observation, as in an ideal sensor (in which the "long lifetime conformation" is the same whether the sensor is bound to the analyte or not, and similarly for the short lifetime one) those values should be identical. While it is entirely possible that this is not the case for G-CaFLITS, since the authors have conducted a calibration experiment using time-domain FLIM, could they show the behavior of the lifetimes and preamplitudes? Are the trends consistent with their interpretation of a different calcium level in the two mitochondrial populations?

We have analyzed the calibration data from TCSPC experiments done with the Leica Stellaris. From these data (acquired at high photon counts as it is purified protein in solution), we infer that both the short and long lifetime do change as a function of calcium concentration. In particular the long lifetime shows a substantial change, which we cannot explain at this moment. We agree that this is interesting and may potentially give insight in the conformation changes that give rise to the lifetime change.

The lifetime data of the mitochondria has been acquired with a different FLIM setup, but the trend is consistent, both the long and short lifetime decrease in the peripheral mitochondria that have a higher calcium concentration.

**Author response image 1. sa4fig1:** 

(6) "The lifetime response of Tq-Ca-FLITS and the ΔF/F response of jGCaMP7f resembled each other, with both signals gradually increasing over the span of 3-4 minutes after we increased external [K+]; the two signals then hit a plateau for ~1 min, followed by a return to baseline and often additional plateaus (Figure 8B-C). By comparison, G-Ca-FLITS responses were more variable, typically exhibiting a smaller ramping phase and seconds-long spikes of activity rather than minutes-long plateaus (Figure 8C)."

This statement does not appear fully consistent with the data in Figure 8. While in figure 8B it looks like GCaMP and mTq-CaFLITS have very similar profiles, these curves come from one single experiment out of a very variable dataset (see Figure 8C). If one would for example choose the second curve of GCaMP in Figure 8C, it would look very similar to the response of G-CaFLITS in figure 8B, and the argument would be reversed. How do the averages look like?

Indeed, the dynamics of the responses are very variable and we do not want to draw attention to these differences in the dynamics, so we have removed the comparison. Instead, the difference in intensity change and lifetime contrast are of importance here. To answer the question of the reviewer, we have added a new panel (D) which shows the average responses for each of the GECIs.

(7) "Although the calibration is equipment independent under ideal conditions, and only needs to be performed once, we prefer to repeat the calibration for different setups to account for differences in temperature or pulse frequency."While I generally agree with the statement, it is imprecise. A change in temperature is generally expected to affect the Kd, so rather than "preferring to repeat", it is a requirement for accurate quantification at different concentrations. I am not sure I understand what the pulse frequency is in this context, and how it affects the Kd.

We thank the referee for pointing out that our text is imprecise and confusing. What we meant to say is that we see differences between different set-ups and we have clarified this by changing the text. We have also added that it is “necessary” to repeat the calibration:

“Although the calibration is equipment independent under ideal conditions, and only needs to be performed once, we do see differences between different set-ups. Therefore, it is necessary to repeat the calibration for different set-ups.”

(8) "A recent effort to generate a green emitting lifetime biosensor used a GFP variant as a template (Koveal et al., 2022), and the resulting biosensor was pH sensitive in the physiological range. On the other hand, biosensors with a CFP-like chromophore are largely pH insensitive (van der Linden et al., 2021; Zhong et al., 2024)."The dismissal of the use of T-Sapphire as a pH independent template is inaccurate. The same group has previously reported other sensors (SweetieTS for glucose and Peredox for redox ratio) that are not pH sensitive. Furthermore, in Koveal et al. also many of the mTq2-based variants showed a pH response, suggesting that the pHdependence for the Lilac sensor might be more complex. Still, G-CaFLITS present advantages in terms of the possibility to excite at longer wavelengths, which could be mentioned instead.

We only want to make the point that adding the T203Y mutation to Turquoise-based lifetime biosensors may be a good approach for generating pH insensitive green biosensors. There is no point in dismissing other green biosensors and we have changed the text to: “Since biosensors with a CFP-like chromophore are largely pH insensitive (van der Linden et al., 2021; Zhong et al., 2024), and we show here that the pH independence is retained for the Green Ca-FLITS, we expect that adding the T203Y mutation to a cyan sensor is a good approach for generating pH-insensitive green lifetime-based sensors.”

(9) "Usually, a higher QY results in a higher intensity; however, in G-Ca-FLITS the open state has a differential shaped excitation spectrum which leads to a decreased intensity. These effects combined have resulted in a sensor where the two different states have a similar intensity despite displaying a large QY and lifetime contrast."This statement does not seem to reflect the excitation spectra of Figure 1. If this explanation would be true, wouldn't there be an isoemissive point in the excitation spectrum (i.e. an excitation wavelength at which emission intensity would not change)?

The excitation spectra in figure 1 are not ideal for the interpretation as these are not normalized. The normalized spectra are shown in figure S10, but for clarity we show the normalized spectra here below as well. For the FD-FLIM experiments we used a 446 nm LED that excites the calcium bound state more efficiently. Therefore, the lower brightness due to a lower QY of the calcium bound state is compensated by increased excitation. So the limited change in intensity is excitation wavelength dependent. We have added a sentence to the discussion to stress this:

“The smallest intensity change is obtained when the calcium-bound state is preferably excited (i.e. near 450 nm) and the effect is less pronounced when the probe is excited near its peak at 474 nm”

(10) "We evaluated the use of Tq-Ca-FLITS and G-Ca-FLITS for 2P-FLIM and observed a surprisingly low brightness of the green variant in an intact fly brain. This result is consistent with a study finding that red-shifted fluorescent-protein variants that are much brighter under one-photon excitation are, surprisingly, dimmer than their blue cousins in multi-photon microscopy (Molina et al., 2017). The responses of both probes were in line with their properties in single photon FLIM, but given the low brightness of G-Ca-FLITS under 2-photon excitation, the Tq-Ca-FLITS may be a better choice for 2P-FLIM experiments."The differences appear strikingly high, and it seems improbable that a reduction in two-photon absorption coefficient might be the sole cause. How can the authors rule out a problem in expression (possibly organism-specific)?

The reviewers are correct that the changes in brightness between G-Ca-FLITS and Tq-Ca-FLITS may arise from changes in expression levels. It is difficult to calibrate for these changes explicitly without a stable reference fluorophore. However, both the G-Ca-FLITS and Tq-Ca-FLITS transgenic flies produced used the same plasmid backbone (the Janelia 20x-UAS-IVS plasmid), landed in the same insertion site (VK00005) of the same genetic background and were crossed to the same Janelia driver line (R60D05-Gal4), so at the level of the transcriptional machinery or genetic regulatory landscape the two lines are probably identical except for the few base pair differences between the G-Ca-FLITS and Tq-Ca-FLITS sequence. But the same level of transcription may not correspond to the same amount of stable protein in the ellipsoid body. So, we cannot rule out any organism-specific problems in expression. To examine the 2P excitation efficiency relative to 1P excitation efficiency, we have measured the fluorescence intensity of purified G-Ca-FLITS and Tq-Ca-FLITS on beads. See also response to reviewer 3 and supplemental figure S14

Suggestions(1) The underlying assumption of any experiment using a biosensor is that the concentration of the biosensor should be roughly 2 orders of magnitude lower than the concentration of the analyte, otherwise the calibration equations do not hold. When measuring nM concentrations of calcium, this problem can be in principle very significant, as the concentration of the sensor in cells is likely in the low micromolar range. Calcium regulation by the cell should compensate for the problem, and the equations should hold. However, this might not hold true during experimental conditions that would disrupt this tight regulation. It might be a good thing to add a sentence to inform users about the limitations in interpreting calcium concentration data under such conditions.

Good point. We have added this to the discussion: “All calcium indicators also act as buffers, and this limits the accuracy of the absolute measurements, especially for the lower calcium concentrations (Rose et al., 2014), as the expression of the biosensor is usually in the low micromolar range.”

(2) Different methods of lifetime "averaging", such as intensity or amplitude-weighted lifetime in time domain FLIM or phase and modulation in frequency domain might lead to different Kd in the same calibration experiment. This is an underappreciated factor that might lead to errors by users. Since the authors conducted calibrations using both frequency and time-domain, it would be useful to mention this fact and maybe add a table in the Supporting Information with the minima, maxima and Kds calculated using different lifetime averaging methods.

To avoid biases due to fitting we prefer to use the phasor plot, this can be used for both frequency and time-domain methods and we added a sentence to the discussion to highlight this: “We prefer to use the phasor analysis (which can be used for both frequency- and time-domain FLIM), as it makes no assumptions about the underlying decay kinetics.”

(3) The origin of the redshift observed in G-CaFLITS is likely pi-stacking, similar to the EGFP-to-EYFP case. While previous studies suggest that for mTq2 based sensors a change in rigidity would lead to a change in the non-radiative rate, which would result in similar changes in quantum yield and (amplitude-weighted average) lifetime. If pi-stacking plays a role, there could be an additional change in the radiative rate (as suggested also by the change in absorption spectra). Could this play a role in the relation between brightness and lifetime in G-CaFLITS? Given the extensive data collected by the authors, it should be possible to comment on these mechanistical aspects, which would be useful to guide future design.

We do appreciate this suggestion, but we currently do not have the data to answer this question. The inverted response that we observe, solely due to the introduction of the tyrosine is puzzling. Perhaps introduction of the mutation that causes the redshift in other cyan probes will provide more insight.

**Reviewer #2 (Recommendations for the authors):**
Specific points:The first section of Results is basically a description of how they chose the lysis conditions for screening in bacteria. I didn't see anything particularly novel or interesting about this, anyone working with protein expression in bacteria likely needs to optimize growth, lysis, purification, etc. This section should be moved to the Methods.

As reviewer 1 lists the thorough documentation of this approach as one of the strengths, we prefer to keep it like this. We see this section as method development, rather than purely a method. When this section would be moved to methods, it remains largely invisible and we think that’s a shame. Readers that are not interested can easily skip this section.

In the Results section Characterization of G-Ca-FLITS, the authors state "Here, the calcium affinity was KD = 339 nM, higher compared to the calibration at 37{degree sign}C. This is in line with the notion that binding strength generally increases with decreasing temperature." However, the opposite appears to be true - at 37C they measured a KD of 209 nM which would represent higher binding strength at higher temperature.

Thanks for catching this, we’ve made a mistake. We rephrase this to “higher compared to the calibration at 37 °C. This is unexpected as it not in line with the notion that binding strength generally increases with decreasing temperature.”

In Figure 8c, there should be a visual indicator showing the onset of application of high potassium, as there is in 8b.

This is a good suggestion; a grey box is added to indicates time when high K+ saline was perfused.

**Reviewer #3 (Recommendations for the authors):**
I think the science of the manuscript is sound and the presentation is logical and clear. I have some stylistic recommendations.Supp Fig 1: The figure requires a bit of "eyeballing" to decide which conditions are best, and figuring out which spectra matched the final conditions took a little effort. Is there a way to quantify the fluorescence yield to better show why the one set of conditions was chosen? If it was subjective, then at least highlight the final conditions with a box around the spectra, making it a different colour, or something to make it stand out.

Thanks for the comment; we added a green box.

Supp Fig 3: Similar suggestion. Highlight the final variant that was carried forward (T203Y). The subtle differences in spectra are hard to discern when they are presented separately. How would it look if they were plotted all on one graph? Or if each mutant were presented as a point on a graph of Peak Em vs Peak Ex? Would T203Y be in the top right?

We have added a light blue box for reference to make the differences clearer.

Supp Fig 4 & Fig 1: Too much of the graph show the uninteresting tails of the spectra and condenses the interesting part. Plotting from 400 nm to 600 nm would be more informative.

We appreciate the suggestion but disagree. We prefer to show the spectra in its entirety, including the tails. The data will be available so other plots can be made by anyone.

Fig 3a: People who are not experts in lifetime analysis are probably not very familiar with the phase/modulation polar plot. There should be an additional sentence or two in the main text that _briefly_ describes the basis for making the polar plot and the transformation to the fractional saturation plot in 3B. I can't think of a good way to transform Eq 3 from Supp Info into a sentence, but that's what I think is needed to make this transformation clearer.

We appreciate the suggestion and feel that it is well explained here:

"The two extreme values (zero calcium and 39 μM free calcium) are located on different coordinates in the polar plot and all intermediate concentrations are located on a straight line between these two extremes. Based on the position in the polar plot, we determined the fraction of sensor in the calcium-bound state, while considering the intensity contribution of both states"

Fig 4: The figure is great, and I love the comparison of different calcium sensors. But where is Tq-Ca-FLITS? I get that this is a figure of green calcium sensors, but it would be nice to see Tq-Ca-FLITS in there as well. The G-Ca-FLITS is compared to Tq-Ca-FLITS in Fig 5. Maybe I'm just missing why the bottom panel of Fig 5 cannot be replotted and included in Fig 4.

The point is that we compare all the data with identical filter sets, i.e. for green FPs.using these ex/em settings, the Tq probe would seriously underperform. Note that the data in fig. 5 is not normalized to a reference RFP and can therefore not be compared with data presented in figure 4.

Fig 6: The BOEC data could easily be moved to Supp Figs. It doesn't contribute much relevant info.

We are not keen of moving data to supplemental, as too often the supplemental data is ignored. Moreover, we think that the BOEC data is valuable (as BOEC are primary cells and therefore a good model of a healthy human cell) and deserves a place in the main manuscript.

2P FLIM / Fig 8 / Fig S4: The lack of brightness of G-Ca-FLITS in the 2P FLIM of fruit fly brain could have been predicted with a 2P cross section of the purified protein. If the equipment to perform such measurements is available, it could be incorporated into Fig S4.

Unfortunately, we do not have access to equipment that measures the 2P cross section. As an alternative, we compared the 2P excitation efficiency with 1P excitation efficiency. To this end, we have used beads that were loaded with purified G-Ca-FLITS or Tq-Ca-FLITS. We have evaluated the fluorescence intensity of the beads using 1P (460 nm) and 2P (920 nm) excitation. Although the absolute intensity cannot be compared (the G-Ca-FLITS beads have a lower protein concentration), we can compare the relative intensities when changing from 1P to 2P. The 2P excitation efficiency of G-Ca-FLITS is comparable (if not better) to that of Tq-Ca-FLITS. This excludes the option that the G-Ca-FLITS has poor 2P excitability. We will include this data as figure S12.

We also have added text to the results: “We evaluated the relative brightness of purified Tq-Ca-FLITS and G-Ca-FLITS on beads by either 1-Photon Excitation (1PE) (at 460 nm) or 2-Photon Excitation (2PE) (at 920 nm) and observed a similar brightness between the two modes of excitations (figure S14). This shows that the two probes have similar efficiencies in 2PE and suggest that the low brightness of GCa-FLITS in Drosophila is due to lower expression or poor folding.” and discussion: “The responses of both probes were in line with their properties in single photon FLIM, but given the low brightness of G-Ca-FLITS under 2-photon excitation in Drosphila, the Tq-Ca-FLITS is a better choice in this system. Yet, the brightness of G-Ca-FLITS with 2PE at 920 nm is comparable to Tq-Ca-FLITS, so we expect that 2P-FLIM with G-Ca-FLITS is possible in tissues that express it well.”